# DRESSing Up LLM: Efficient Stylized Question-Answering via Style Subspace Editing

**Xinyu Ma**[1], **Yifeng Xu**[1], **Yang Lin**[1], **Tianlong Wang**[3], **Xu Chu**[1,2,3], **Xin Gao**[1],
**Junfeng Zhao**[1], **Yasha Wang**[1,3]*
[1] School of Computer Science, Peking University
[2] Center on Frontiers of Computing Studies, Peking University
[3] National Research and Engineering Center of Software Engineering, Peking University
{maxinyu,wangyasha}@pku.edu.cn

## Abstract

We introduce DRESS, a novel approach for generating stylized large language model (LLM) responses through representation editing. Existing methods like prompting and fine-tuning are either insufficient for complex style adaptation or computationally expensive, particularly in tasks like NPC creation or character role-playing. Our approach leverages the over-parameterized nature of LLMs to disentangle a style-relevant subspace within the model's representation space to conduct representation editing, ensuring a minimal impact on the original semantics. By applying adaptive editing strengths, we dynamically adjust the steering vectors in the style subspace to maintain both stylistic fidelity and semantic integrity. We develop two stylized QA benchmark datasets to validate the effectiveness of DRESS, and the results demonstrate significant improvements compared to baseline methods such as prompting and ITI. In short, DRESS is a lightweight, train-free solution for enhancing LLMs with flexible and effective style control, making it particularly useful for developing stylized conversational agents. [1]

## 1 Introduction

Large language models (LLMs) like GPT-4 (Achiam et al., 2023) and LLaMA-3 (Dubey et al., 2024) have demonstrated exceptional performance across a range of natural language processing (NLP) tasks including question-answering. This evokes the wide use of LLMs as conversational agents (Weizenbaum, 1966) for various applications, including psychological counseling (Li et al., 2023a), creating gaming NPCs (non-player characters) (Cox & Ooi, 2023) and character simulacra (Shao et al., 2023). While LLMs are adept at providing accurate and coherent answers, they lack the intrinsic ability to tailor responses in a specific ***language style***. Language style (Jin et al., 2022) is linguistically defined as the manner of expressing the semantics, depicted by multiple attributes like personality, emotion, authorship, era

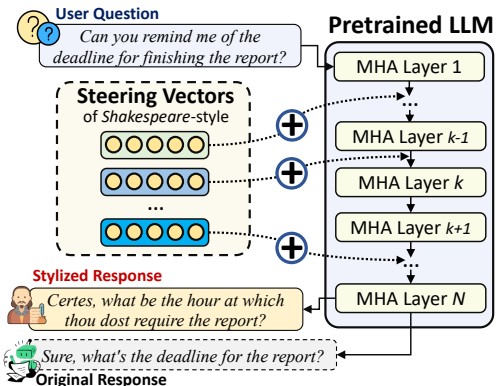

Figure 1: An illustrative example of representation editing for *Shakespeare*-style responses.

background, etc. Stylized responses are crucial for LLM agents as the style can shape the interaction tone, making the agents more immersive and engaging, and ensuring that responses are empathetic and appropriately tailored to the user's emotional states. Hence, crafting the language style is essential for shaping the specific image and personality of conversational agents. Therefore, we aim to solve the following question: ***How to make LLMs respond to user questions in a specific style?***

Currently, there are two main approaches to achieving stylized responses - prompting with few-shot demonstrations and fine-tuning. Prompting methods (Park et al., 2023) leverage the in-context

---

*Corresponding Author
[1]Codes and benchmark datasets are available at https://github.com/ArthurLeoM/DRESS-LLM.

learning ability (Brown et al., 2020) of LLMs by using a description of the target style along with few-shot examples to generate stylized responses. However, simply prompting LLMs is no longer proper as instructions are plain and insufficient to describe a certain style comprehensively, and demonstrations could severely increase the sequence length, increasing the risk of lost-in-the-middle (Liu et al., 2024). A better way is to conduct supervised fine-tuning (SFT) (Ma et al., 2024) with target style response data (Shao et al., 2023), where LLM's outputs are adapted to the target style distribution by adjusting the model parameters. Yet this approach is overly burdensome, particularly for scenarios like game NPC construction. Each character requires a separate fine-tuning process, making the creation of multiple characters extremely costly in terms of time and computational resources. Therefore, it is necessary to develop an **effective** and **efficient** strategy to reach our goal.

Representation editing (Burns et al., 2023; Turner et al., 2023) has recently been widely used to control specific behaviors of LLMs (e.g., truthfulness enhancement (Zou et al., 2023), knowledge editing (Hernandez et al., 2023), etc.). Since it operates solely on the representation space without optimizing the parameter space, it is **lightweight, train-free, and highly efficient**. Additionally, it leverages large amounts of data to compute generalizable steering vectors for depicting specific model functions, making it highly **effective**. Building on this insight, our approach attempts to utilize representation editing methods to craft the style of LLM output. Specifically, as shown in Figure 1, we aim to solve a steering vector that is added to LLM's activations during inference, shifting the representations to the direction of another language style (e.g., poetic and rhythmic Shakespearean early modern English). This approach fulfills our need to combine the efficiency of a train-free method with the effectiveness of data-driven steering for stylizing LLM responses.

However, when building stylized conversational agents, it is also crucial to ensure the response quality alongside stylization. In other words, generating stylized responses must not compromise the original semantics. This presents a significant technical challenge for our representation editing approach: ***How to solve a steering vector minimizing the influence on the underlying semantics?*** Recent research observes that in the extremely wide and high-dimensional space of over-parameterized LLMs, activations can be assumed to be approximately orthogonal with high probability (Wang & Zhu, 2023). This implies that the different language functions are likely to reside in orthogonal and disentangled linear subspaces (Ortiz-Jimenez et al., 2023). Hence, our insight is to identify a style-relevant and semantic-isolated subspace from the representation space to edit within. Building on this insight, we propose DRESS (**D**isentangling **R**epresentation **E**diting in **S**tyle **S**ubspace), comprising the following strategies to progressively locate the style subspaces and perform semantic-isolated style steering. **1) Attention head filtering**: It has been demonstrated that different attention heads tend to perform varying functions (Ge et al., 2024). Hence we use probing techniques to identify the attention heads that are more closely related to styles and edit within those heads. **2) Style subspace filtering**: To further eliminate the style-irrelevant components in the selected attention heads, we conduct subspace filtering by seeking a subspace supported by style-related bases, so that the impact on semantics could be minimized. **3) Adaptive editing strength**: We employ adaptive editing strength on each subspace basis and each generated token to provide higher flexibility and avoid excessively intense editions that could harm the semantics. Compared to previous methods (Zou et al., 2023; Li et al., 2023b) relying on a single steering vector for editing, our approach offers greater flexibility and expressiveness by introducing a higher-rank subspace to represent style. Meanwhile, it filters out style-irrelevant noises within the steering vector, allowing for better semantic preservation.

To validate the effectiveness of our approach, we construct an evaluation benchmark comprising two specific stylized question-answering datasets of different languages (i.e., *Shakespeare*-style in English and *Dream of the Red Chamber*-style in Chinese[2]). The objective evaluation metrics include style intensity, semantic preservation, and fluency, following traditional criteria (Jin et al., 2022, Section 3). Additionally, we utilize the GPT-4 rating as a surrogate for human evaluation (Zheng et al., 2023), serving as an overall assessment metric to comprehensively evaluate the model's capabilities.

To summarize, we highlight our contributions as follows. We proposed a lightweight and train-free representation editing method dubbed DRESS based on the decoupling of language style subspaces to enable stylized LLM QA systems, which lays a fundamental groundwork for constructing humanoid conversational agents. Technically, we propose three mechanisms to progressively isolate the style-relevant subspace from the entire representation space, improving the expressiveness of the style

---

[2]Here we select English, the most widely used language globally, and Chinese, the language with the largest number of native speakers, as our two examples. For dataset details, please refer to Section 5.1.

and ensuring that the semantics of LLMs remain unaffected. Finally, we introduced a benchmark to evaluate the response quality of stylized QA. DRESS shows significant improvements over SOTA baselines, including SFT, prompting, and other representation editing methods, demonstrating the effectiveness of our method.

## 2  RELATED WORKS

Recently, there has been a line of research embarking on controlling the behavior of LLMs through representation editing, most of which focuses on truthfulness enhancement (Zou et al., 2023; Li et al., 2023b), knowledge editing (Todd et al., 2023; Hernandez et al., 2023), etc. This technique is based on the linear representation hypothesis (Elhage et al., 2022) supposing that most high-level concepts are represented linearly as directions in LLMs, which is theoretically supported by the approximate orthogonality assumption under overparameterized networks (Wang & Zhu, 2023), and practically demonstrated by the success of linear probing techniques (Alain & Yoshua, 2016; Belinkov, 2022).

The primary objective of representation editing is to identify some steering vectors and add them to some layers of the forward pass of LLMs to introduce certain attributes (i.e., language style in this work or truthfulness, etc.) into the LLM outputs. Mean-Centring (Jorgensen et al., 2023) computes the steering directions using the mean difference between paired activations. RepE (Zou et al., 2023, Representation Engineering) applies PCA to the set of difference vectors and selects the principal component as the steering vector. CCS (Burns et al., 2023, Contrast Consistence Search) obtains the steering vector through the probing vector that well classifies the activation pairs. ITI (Li et al., 2023b, Inference-Time Intervention) further enhances CCS by locating attribute-relevant attention heads. However, due to the intricacy of language attributes, it is insufficient to depict them with a single direction as in the aforementioned works. TrFr (Chen et al., 2024b, Truth Forest) proposes a specific combination of several vectors under orthogonality regularization to enhance the expressiveness of the target attribute. Nevertheless, none of the methods above attempt to explicitly disentangle the attribute subspace from the entire representation space to avoid affecting the original semantics. Moreover, previous works overlook the varying importance of different attribute components across various contexts, which can adversely affect the quality of the outputs. In this work, we propose DRESS to solve the problems. DRESS comprises three progressive mechanisms to isolate the attribute-relevant subspace and conduct adaptive editing in order to enhance the expressiveness and flexibility of steering, meanwhile ensuring the semantics are preserved.

## 3  PRELIMINARIES

**Problem Formulation**    In this paper, we aim at making LLMs respond to user queries in a specific style. Rigorously, given each user query $\boldsymbol{q}$, an LLM $\boldsymbol{M}(\cdot)$ to respond the query with $\boldsymbol{M}(\boldsymbol{q})$ as the original response, and a target language style $\mathcal{S}$ depicted by QA examples $\{\boldsymbol{q}_i, \boldsymbol{a}_i\}_{i=1}^n$ where $\boldsymbol{a}_i$ are all stylized responses (i.e., $\boldsymbol{a}_i \sim \mathcal{S}$), our objective is to edit the representation space of LLM and obtain a new response $\boldsymbol{M}'(\boldsymbol{q})$ of user query $\boldsymbol{q}$, where the response $\boldsymbol{M}'(\boldsymbol{q})$ is of the same style with $\mathcal{S}$ (i.e., $\boldsymbol{M}'(\boldsymbol{q}) \sim \mathcal{S}$).

**Representation Editing**    Here we rigorously introduce where representation editing takes place in the transformer-based LLMs. To set notation and contexts, we first briefly introduce the transformer (Vaswani, 2017) architecture adopted by mainstream LLMs. A transformer-based LLM comprises several stacked transformer blocks, each composed of a multi-head self-attention (MHA) block and a successive MLP layer. Specifically, a transformer block could be expressed as follows:

$$\boldsymbol{x}^{(l+1)} = \mathrm{MLP}(\mathrm{MHA}(\boldsymbol{x}^{(l)})) = \mathrm{MLP}(\bigoplus_{h=1}^{H} \mathbf{W}_h^o(\mathrm{Attn}_h(\boldsymbol{x}^{(l)}))). \tag{1}$$

It has been demonstrated that the MHA block and the feed-forward network (FFN) perform different functions in LLM, where MHA blocks tend to encode language attributes (Clark, 2019) while FFNs tend to conduct reasoning (Geva et al., 2020). Hence, it is more reasonable to edit representations in MHA blocks to minimize the influence on semantics. Specifically, the edited steering vector is attached after the $\mathrm{Attn}$ operator and before $\mathbf{W}^o$ following Li et al. (2023b); Chen et al. (2024b):

$$\tilde{\boldsymbol{x}}^{(l+1)} = \mathrm{MLP}(\mathrm{MHA'}(\boldsymbol{x}^{(l)})) = \mathrm{MLP}(\bigoplus_{h=1}^{H} \mathbf{W}_h^o(\mathrm{Attn}^h(\boldsymbol{x}^{(l)}) + \boldsymbol{v}^{(h,l)})), \tag{2}$$

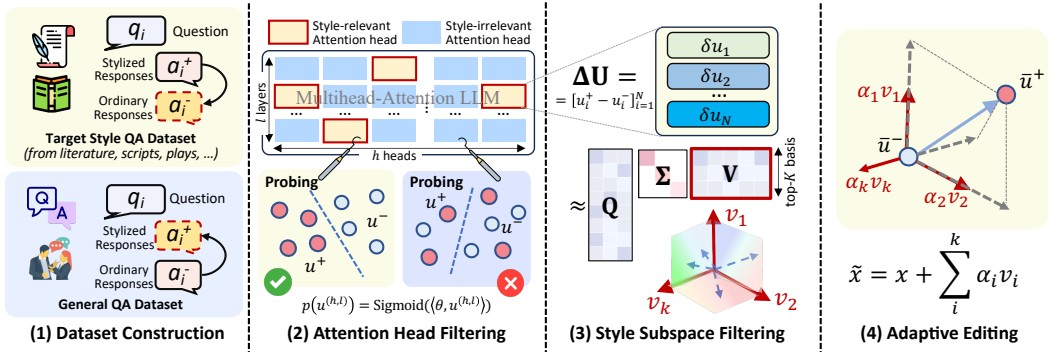

Figure 2: The overall pipeline of DRESS. We first process the target-style QA dataset into a form suitable for solving the steering vector. Next, we use probes to filter out the attention heads most relevant to the style and further disentangle the style-related subspaces within the representation space of these heads, where the steering vectors are computed. Finally, during editing, we apply an adaptive editing strength mechanism to control the magnitude of different sub-directions in the style subspace, optimizing the editing quality while avoiding negative impacts on the output semantics.

where $\boldsymbol{v}^{(h,l)} \in \mathbb{R}^d$ is the steering vector to be solved for editing the $h$-th head in $l$-th layer, and we denote $\boldsymbol{u}^{(h,l)} \in \mathbb{R}^d$ as the original activation of $h$-th head in $l$-th layer, i.e., $\boldsymbol{u}^{(h,l)} = \text{Attn}^h(\boldsymbol{x}^{(l)})$.

## 4 METHODS

In this section, we introduce how DRESS solves the steering vectors and conducts representation editing for stylized outputs without compromising the semantics. Specifically, the pipeline is shown in Fig.2, and we introduce the details as follows.

### 4.1 DATASET CONSTRUCTION

To conduct effective representation editing, it is necessary to investigate the differences between the activations of QA samples with different styles but the same semantics for deriving a style-relevant steering vector. The target style is inherently implied by QA examples $\{\boldsymbol{q}_i, \boldsymbol{a}_i\}_{i=1}^n$, which are collected from literature, scripts, or chat records. Therefore, to compute the steering vector, we also need to obtain the ordinary style of these responses (i.e., the style LLM generates), thereby constructing the dataset $\mathcal{D} = \{\boldsymbol{q}_i, \boldsymbol{a}_i^-, \boldsymbol{a}_i^+\}_{i=1}^n$ to solve the steering vector, where $\boldsymbol{a}_i^-$ is the response to $\boldsymbol{q}_i$ in the ordinary style and $\boldsymbol{a}_i^+$ is the collected target style response. To obtain the ordinary style expression of $\boldsymbol{a}_i^+$ (i.e., $\boldsymbol{a}_i^-$) without altering its semantics, we apply GPT-4 to rewrite $\boldsymbol{a}_i^+$ to align with the typical LLM language style (i.e., modern daily language style). The specific prompt used for this task can be found in Appendix C.1.

Additionally, since the dataset often originates from scripts and literary works, the language style of the queries tends to be biased. To mitigate the influence, we introduce another general-purpose LLM QA dataset (e.g., Alpaca (Taori et al., 2023), MOSS (Sun et al., 2024)) $\mathcal{D}' = \{\boldsymbol{q}_i', \boldsymbol{a}_i^{-\prime}, \boldsymbol{a}_i^{+\prime}\}_{i=1}^{n'}$, to diversify the style distribution of the queries. Specifically, the general-purpose QA dataset already contains the ordinary style QA data pair (i.e., $\boldsymbol{q}_i', \boldsymbol{a}_i^{-\prime}$), so we need to construct corresponding target style responses $\boldsymbol{a}_i^{+\prime}$ to perform data augmentation. Here, we again prompt GPT-4 to generate the target style responses, with a brief introduction of the target style and randomly sampled target style responses $\boldsymbol{a}_i^+$ from the collected dataset $\mathcal{D}$ as few-shot examples. The detailed prompt can be found in Appendix C.2. Finally, the dataset are constructed as $\mathcal{D} := \mathcal{D} \cup \mathcal{D}'$ sized $N = n + n'$.

### 4.2 ATTENTION HEAD FILTERING

Recent works (Ge et al., 2024) have demonstrated that different attention heads perform different functions in LLMs. Therefore, identifying the attention heads most closely related to styles is crucial for conducting semantic-isolated representation editing. Probing, as highlighted in works like (Alain & Yoshua, 2016; Conneau et al., 2018; Belinkov, 2022), has emerged as a robust and effective technique for analyzing the internal functions and behavior patterns within LLM representations. Our key idea is to train a linear probing classifier on the activations of LLMs to discriminate between the ordinary and target language styles. Since each pair of responses in our dataset (i.e., $\boldsymbol{a}_i^-, \boldsymbol{a}_i^+$)

shares the same semantics but only differs in style, we can determine whether an attention head is style-relevant based on the probing accuracy of the style classification task.

Hence, in DRESS , we define the probe $p(\boldsymbol{u}^{(h,l)}) = \text{Sigmoid}(\langle \boldsymbol{\theta}, \boldsymbol{u}^{(h,l)} \rangle)$ for each head $h$ in each layer $l$ of the LLM to detect the style-relevance of the activations. For each sample, we concatenate the queries $\boldsymbol{q}_i$ and responses $\boldsymbol{a}_i$ and extract the activations at the last token, where the semantics are completely encoded and ensured to be the same for each pair of $\boldsymbol{a}_i^-$ and $\boldsymbol{a}_i^+$. Then we create a probing dataset $\mathcal{D}_h^{(l)} = \{(\boldsymbol{u}_i^{(h,l)}, y_i)\}$ for each head in each layer, where $y$ indicates whether the current activation originates from the ordinary or target style. Specifically,

$$\mathcal{D}_h^{(l)} = \left\{ \left( \boldsymbol{M}(\boldsymbol{q}_i, \boldsymbol{a}_i^+)^{(h,l)}, y^+ \right) \right\}_{i=1}^N \cup \left\{ \left( \boldsymbol{M}(\boldsymbol{q}_i, \boldsymbol{a}_i^-)^{(h,l)}, y^- \right) \right\}_{i=1}^N, y^+ = 1, y^- = 0. \quad (3)$$

Next, we randomly split each dataset into training and validation sets in a 4:1 ratio, fitting the binary linear classifier $p(\cdot)$ on the training set. We select the attention heads with the top-$H$ validation accuracy as style-relevant and conduct editing within those heads.

### 4.3 STYLE SUBSPACE FILTERING

Given the selected attention heads, we aim to further filter out the style-irrelevant components and disentangle the subspaces that are more closely related to style for editing. Since the activations in the high-dimensional space of LLMs can be assumed to be approximately orthogonal with high probability (Wang & Zhu, 2023; Ortiz-Jimenez et al., 2023), we can hypothesize that the language styles reside in a subspace orthogonal with semantics. Given that our positive and negative sample pairs (i.e., $(\boldsymbol{q}_i, \boldsymbol{a}_i^-), (\boldsymbol{q}_i, \boldsymbol{a}_i^+)$) differ only in style while maintaining consistent semantics, their activation differences (i.e., $\delta\boldsymbol{u}_i^{(h,l)} = \boldsymbol{u}_i^{(h,l)+} - \boldsymbol{u}_i^{(h,l)-}$) primarily capture the variation in style, with minimal inclusion of semantic or other noisy components. Thus, DRESS proposes to isolate the style-relevant subspace by denoising the space spanned by these activation differences.

Specifically, we first collect the activation differences of all sample pairs, denoted as $\Delta\mathbf{U}^{(h,l)} = [\delta\boldsymbol{u}_1^{(h,l)}, \delta\boldsymbol{u}_2^{(h,l)}, \cdots, \delta\boldsymbol{u}_N^{(h,l)}]^\top \in \mathbb{R}^{N \times d}$. Then we apply Singular Value Decomposition (SVD) on $\Delta\mathbf{U}^{(h,l)}$, and select the top-$K$ singular vectors with the largest singular values to form the orthogonal basis of the style subspace, thereby capturing the most representative style-related features while filtering out irrelevant noises. Rigorously,

$$\Delta\mathbf{U}^{(h,l)} = \mathbf{S}^{(h,l)} \boldsymbol{\Sigma}^{(h,l)} \mathbf{V}^{(h,l)\top} = \sum_{i=1}^d \sigma_i \boldsymbol{s}_i^{(h,l)} \boldsymbol{v}_i^{(h,l)} \approx \sum_{i=1}^K \sigma_i \boldsymbol{s}_i^{(h,l)} \boldsymbol{v}_i^{(h,l)}, \quad (4)$$

where $\boldsymbol{v}_i^{(h,l)} \in \mathbb{R}^d$ is the singular vector and $\sigma_i \in \mathbb{R}$ is the corresponding singular value satisfying $\forall i > j, \sigma_i > \sigma_j$. Finally, the editing is conducted in the style subspace spanned by $\boldsymbol{v}_i^{(h,l)}$ as follows:

$$\tilde{\boldsymbol{x}}^{(l+1)} = \text{MLP}(\bigoplus_{h=1}^H \mathbf{W}_h^o (\text{Attn}^h(\boldsymbol{x}^{(l)}) + \sum_{i=1}^K \alpha_i^{(h,l)} \boldsymbol{v}_i^{(h,l)})), \quad (5)$$

where $\alpha_i^{(h,l)}$ is the editing strength of the corresponding basis $\boldsymbol{v}_i^{(h,l)}$ in the style subspace, and especially, for attention heads that have been filtered out in the previous step, $\alpha_i^{(h,l)} = 0$.

### 4.4 ADAPTIVE EDITING

Since different style components (e.g., tone, formality) may have varying importance or influence depending on the specific context, a uniform adjustment would fail to capture these subtleties. Thus, in this subsection, we introduce our adaptive editing strategy, designed with the adaptive strength coefficient $\alpha_i^{(h,l)}$ in Eq.(5). This coefficient comprises two key components: a global editing strength and an adaptive scaling factor. The global editing strength reflects the population-level steering intensity across the dataset, capturing the overall style shift observed in the majority of the samples. Specifically, the global editing strength, denoted as $\beta_i^{(h,l)}$, is measured by the projection of the mean difference between positive and negative activations (i.e., $\overline{\delta\boldsymbol{u}}^{(h,l)} = \frac{1}{N}\sum_{i=1}^N \delta\boldsymbol{u}_i^{(h,l)}$) onto the

orthogonal basis $\boldsymbol{v}_i^{h,l}$ of the style subspace:

$$\beta_i^{(h,l)} = \langle \overline{\delta\boldsymbol{u}}^{(h,l)}, \boldsymbol{v}_i^{h,l} \rangle = \left\| \overline{\delta\boldsymbol{u}}^{(h,l)} \right\| \cos\langle \overline{\delta\boldsymbol{u}}^{(h,l)}, \boldsymbol{v}_i^{h,l} \rangle. \tag{6}$$

The adaptive scaling factor is dynamically determined during the generation of each token on each subspace basis. For each token's current activation $\boldsymbol{u}^{(h,l)}$, we observe the difference between $\boldsymbol{u}^{(h,l)}$ and the mean activations of all target style samples (i.e., $\bar{\boldsymbol{u}}^{(h,l)+} = \frac{1}{N}\sum_{i=1}^{N}\boldsymbol{u}_i^{(h,l)+}$) under the style subspace projection. This projection represents the approximate difference between the current token and the target style, which dictates how much strength we should further attach to each basis and guide the token's activation closer to the target style in a context-appropriate manner, leading to a more accurate and flexible stylization. Specifically, the adaptive scaling factor is designed as follows:

$$\gamma_i^{(h,l)} = \cos\langle (\bar{\boldsymbol{u}}^{(h,l)+} - \boldsymbol{u}^{(h,l)}), \boldsymbol{v}_i^{h,l} \rangle, \tag{7}$$

where $\gamma_i^{(h,l)}$ computes the correlation between the style differences and the corresponding basis, depicting how much strength should be augmented to or derived from the current edition. $\gamma_i^{(h,l)}$ is further attached to the global strength to conduct adaptive steering with $(1 + \gamma_i^{(h,l)})\beta_i^{(h,l)}$. Finally, we introduce a hyperparameter $\lambda$ to control the overall style edition strength:

$$\alpha_i^{(h,l)} = \lambda(1 + \gamma_i^{(h,l)})\beta_i^{(h,l)}. \tag{8}$$

This strategy enables the model to control its editing strength in real-time generation, aligning more closely with the desired style while preserving the integrity of the original content. We also present the algorithmic pseudo-code of DRESS in Appendix A.

## 5 EXPERIMENTS

### 5.1 EVALUATION BENCHMARK

**Datasets** We constructed the evaluation benchmark with representative language styles in Chinese and English, i.e., *Shakespeare*-style and *Dream of the Red Chamber*-style. These styles exhibit significant differences from contemporary language in tone, idiomatic expressions, historical context, etc., making them easy to observe and evaluate. The *Shakespeare*-style benchmark aims to mimic the language style in Shakespeare's works, with the dataset derived from the original texts of his plays[3] following (Xu et al., 2012). The QA pairs are constructed from excerpts of single-round conversations between different characters in the plays. *Dream of the Red Chamber* is a lengthy fictional novel published in the 18th century and is one of China's Four Great Classical Novels. The *Dream of the Red Chamber*-style benchmark aims to replicate the dialogue style of its characters, with the dataset sourced from the original novel and adapted scripts from film and television. Similarly, the QA pairs are constructed from individual character dialogues in these works.

Additionally, as mentioned in Section 4.1, for each dataset, we incorporated the general question-answer dataset (i.e., MOSS (Sun et al., 2024) in the corresponding language) to address the bias in question style distribution. We then randomly divided each of them into training and testing sets at a ratio of 10:1. The training set is used to solve the stylized QA model, while the testing set only utilizes the questions as the test queries to evaluate the model performances. The detailed statistics and the examples of the datasets are introduced in Appendix E.

**Evaluation Metrics** A successful stylized response not only needs to demonstrate the target style, but also ensures that the original semantics are preserved and the language remains fluent given the inherent uncontrollability of LLMs. Hence, following Jin et al. (2022), we evaluate the quality of the stylized responses in three aspects, including style intensity, semantic preservation, and fluency:

- **Style Intensity (SI)**: we leverage a separately trained style classifier to distinguish whether the response could demonstrate the target style (Shen et al., 2017). Specifically, the classifier is fine-tuned on BERT (Devlin et al., 2018) models [4] using the responses of the target style as positive samples and those of the ordinary style as negative samples. The style intensity is calculated as: $\frac{\text{\#. Responses classified as the target style}}{\text{\#. All responses}}$, ranging from $[0, 1]$.

---

[3]https://www.shakespeareswords.com/

[4]We use BERT-base for English dataset and Chinese-BERT-wwm-ext (Cui et al., 2021) for Chinese dataset.

- **Semantic Preservation (SP)**: Semantic preservation aims to reveal whether the stylized responses semantically deviate from the original output. Hence, we apply the averaged cosine similarities between the semantic embeddings of the original and the stylized responses of LLMs (Fu et al., 2018), encoded by BGE (Chen et al., 2024a) embedding model. This score ranges from $[0, 1]$.
- **Fluency Score (FS)**: We also utilize the perplexity metric calculated by the original LLM (i.e., before representation editing) to depict the language fluency. Since perplexity ranges from $[1, \infty)$, differs in an exponential magnitude, and is negatively correlated with fluency, we design the fluency score of a response as $\frac{1}{1+\log \mathrm{PPL}}$. This score ranges from $(0, 1]$ where the values are re-scaled in a more uniform manner, and the higher the score, the more fluent the response. To depict the population-level performance, we report the mean fluency score across all stylized responses.

We also design an objective overall assessment score (OA) using the products of the three metrics (i.e., OA = SI*SP*FS, the higher the better), balancing the trade-off effects between them. Furthermore, we utilize GPT-4 (Achiam et al., 2023) to rate the stylized responses comprehensively, with scores ranging from 0 to 10.[5] The averaged GPT-4 rating is reported for subjective overall assessment.

**Baselines**  We adopt the following state-of-the-art approaches as our compared baselines.

- **Few-shot Prompting** leverages the in-context learning ability to achieve stylized responses. Specifically, we use a well-crafted prompt to describe the target style (See Appendix C.3 for detailed prompts), alongside randomly sampled 3-shot examples from the training set as the demonstrations.
- **Supervised Fine-Tuning (SFT)** incorporates the stylized QA samples in the training set as supervision and tunes the model parameters to adapt the outputs to the target style. Here we apply the state-of-the-art PEFT algorithm LoRA (Hu et al., 2021) as our fine-tuning strategy.
- **Representation Editing** aims to solve generalizable steering vectors attached to LLM internal activations for style editing. Here we include several state-of-the-art representation editing methods as baselines, which have demonstrated superior performances on controlling truthfulness, such as Mean-Centring (Jorgensen et al., 2023), RepE (Zou et al., 2023), ITI (Li et al., 2023b)), and TrFr (Chen et al., 2024b). The details are discussed in Section 2.

**Implementation Details**  We apply Qwen-1.5-14B-Chat (Bai et al., 2023) as our base LLM to experiment on. The experiments are conducted on a machine equipped with 8 NVIDIA-RTX3090-24GB GPUs. All the hyperparameters (e.g., the number of selected attention heads $H$, editing strength $\lambda$, etc.) are tuned via grid search. See Appendix D for more details.

## 5.2 EXPERIMENTAL RESULTS

**Quantitative Analysis**  Table 1 presents the performance of various methods on two stylistic QA evaluation benchmarks. In addition to conventional baseline methods, we also include a comparison with DRESS* as an ablation study, which removes the adaptive scaling factor $\gamma$ during inference, with a fixed editing strength $\alpha_i^{(h,l)} = \lambda \beta_i^{(h,l)}$. It can be observed that our method demonstrates significant performance improvements over all previous approaches, including few-shot prompting, supervised fine-tuning, and all conventional representation editing methods. On *Shakespeare*-style benchmark, DRESS exhibits 7.84% improvements on overall assessment and 1.37% on GPT-4 rating compared to the best-performing baseline. On *Dream of the Red Chamber*-style benchmark, the improvements reach as high as 23.8% on overall assessment and 4.19% on GPT-4 rating, respectively, demonstrating the effectiveness of our method. Below are some key findings:

- **Conventional representation editing methods are not sufficient for stylized QA.** Though demonstrated effective in enhancing LLM truthfulness, most conventional methods cannot reach the performance of few-shot prompting. The performance gap can be attributed to the ignorance of disentangling style from semantic, which can potentially damage the original semantics and even affect the general language ability of LLMs. ITI attempts to locate style-relevant attention heads to edit within, but still suffers from noises underlying the edited representation space. Hence, we think it is crucial to isolate the style subspace for representation editing on stylized QA tasks.

---

[5]Please refer to Appendix C.4 for the rating prompts.

Table 1: Experimental results on two stylized-QA benchmark, *Dream of the Red Chamber*-style and *Shakespeare*-style. The stylized responses are evaluated through style intensity (SI), semantic preservation (SP), fluency score (FS), and overall assessments, including objective assessment (OA = SI*SP*FS) and GPT-4 rating. For all metrics, higher scores indicate better performance. The first and second best-performing methods are respectively highlighted in **bold** and underline.

| Method | *Dream of the Red Chamber*-style (Chinese) | | | | | *Shakespeare*-style (English) | | | | |
|---|---|---|---|---|---|---|---|---|---|---|
| | SI (%) | SP (%) | FS (%) | OA (%) | GPT-4 | SI (%) | SP (%) | FS (%) | OA (%) | GPT-4 |
| Prompt | 93.0 | 66.2 | 36.8 | 22.7 | 7.48 | 98.0 | 69.9 | 37.8 | 25.9 | 8.58 |
| SFT | 85.3 | 69.0 | 40.0 | 23.5 | 7.19 | 95.5 | 69.8 | 36.8 | 24.5 | 8.08 |
| Mean-Centring | 77.5 | 63.6 | 31.4 | 15.5 | 5.63 | 94.5 | 71.5 | 35.3 | 23.9 | 8.26 |
| RepE | 58.5 | 67.7 | 42.2 | 16.7 | 5.64 | 94.5 | 65.2 | 34.3 | 21.1 | 6.91 |
| TrFr | **99.0** | 69.7 | 33.9 | 23.4 | 6.96 | 99.3 | 70.8 | 38.2 | 26.8 | 8.73 |
| ITI | 84.7 | 70.3 | 36.7 | 21.8 | 7.64 | **99.5** | 70.5 | 36.3 | 25.5 | 8.60 |
| DRESS* | 89.0 | **70.9** | 37.8 | 23.8 | 7.71 | 99.0 | 71.2 | 38.2 | 26.9 | 8.63 |
| DRESS | 97.0 | 70.8 | **42.4** | **29.1** | **7.96** | **99.5** | **73.3** | **39.6** | **28.9** | **8.85** |

Table 2: Case study for qualitative analysis. Two cases for *Shakespeare* (***Q1***: general QA, ***Q2***: plot-related QA) and one case for *Dream of the Red Chamber* (***Q3***: plot-related QA) are presented.

| DRESS Response | ITI Response | Unedited LLM Response |
|---|---|---|
| [*Shakespeare*-style] ***Q1***: *Which of these tourist attractions is the most visited?* | | |
| Verily, the statement thou presenteth is incomplete. I require more data, for there are many tourist attractions world wide, each with their own allure. Kindly, furnish me with a land or region, that I may direct my search thereunto. | Verily, I shall not dilate upon that which is superfluous. The question asketh which tourist's joyance is ofthest visited, and thus, without further ado, I imparteth unto thee, that which the folk in mass doth frequent most: The Great Wall of China, or the Colosseum of Rome, or the Parke Royal of olde Londinium. Verily, these be the tripartite, whereof the People's will doth bend its course. | I would need more information about the specific tourist attractions you are referring to in order to determine which one is the most visited. Please provide me with a list or some details about the attractions in question. |
| [*Shakespeare*-style] ***Q2***: *Pardon this fault, and by my soule I sweare I neuer more will breake an oath with thee.?* | | |
| Verily, I do accept thy penance and thy vow, that thou ne'er again breake bond with me. Thy word is as a gem, precious and true, and I do hold it in the hallowed hallows of mine own heart. So, let us proceed in this our discourse, without further let or hindrance. | Verily, thou hast repented of thine ill, and forsooth, I shall no more entreat an oath from thee. Felicite of thy contrition doth solace me. In peace we rest. | Given your oath, I forgive your mistake and hope you will never break my oath again. |
| [*Dream of the Red Chamber*-style] ***Q3***: 你和我少作怪。你知道你琏二爷爱上了你，要和老太太讨了你作小老婆呢。 | | |
| 这话可得慎言。琏二爷是打心里喜欢我，但咱们俩的命不同，他是金凤凰，我是泥巴胎。老太太心里有数，不会让他轻举妄动的。咱们俩还是安分守己的好，别闹出笑话来。 | 这话休要乱说，咱们姐妹们守着些儿，别带累人。琏二爷那等样人，我还不知他么，凭他怎么肯，老太太也不肯的。再者，我这身子也不配他。你可别拿我取笑儿。 | 多谢琏二爷的心意，但人各有命，我心中已有归属，不敢妄想。还是让彼此保持尊重，各自安好吧。 |

- **Adaptive editing enhances the quality of style editing.** Comparing DRESS* with DRESS , we observe significant improvements across all metrics, highlighting the effectiveness of adaptive editing. On the one hand, using adaptive editing strengths for each style basis substantially improves the expressiveness and flexibility in capturing style, thereby optimizing editing quality. On the other hand, context-aware strength adjustment ensures the appropriate intensity for each token, preventing over-editing or under-editing, thereby improving robustness.

- ***Dream of the Red Chamber*-style is a harder benchmark**. The results show that it is not easy to reach a high SI score and GPT-4 rating on this benchmark, and the performance gap is obviously more significant. The difficulty lies in its complex mix of classical Chinese, fewer similar corpus seen during LLM pretraining, and very deep cultural references, which require nuanced understanding beyond language rules. This makes *Dream of the Red Chamber* more challenging to emulate. Hence, in further analyses, we mostly use this challenging task for observation.

**Qualitative Analysis**  We present several QA cases for qualitative analysis in Table 2. We can observe that DRESS can provide responses of significant stylistic language features and meanwhile shows higher consistency with the original response compared with ITI. For instance, both DRESS and unedited LLM avoid listing examples in the response to ***Q1***, whereas ITI attempts to suggest attractions like the Great Wall. In ***Q2***, DRESS provides more metaphors (e.g., *as a gem, precious and true*) and rhythmed arrangements, which is more likely to be in a Shakespearean play. Meanwhile, ITI lost the semantics of *never break my oath* in the original response, while DRESS depicts it with *without further let or hindrance*. For more cases, please refer to Appendix F.

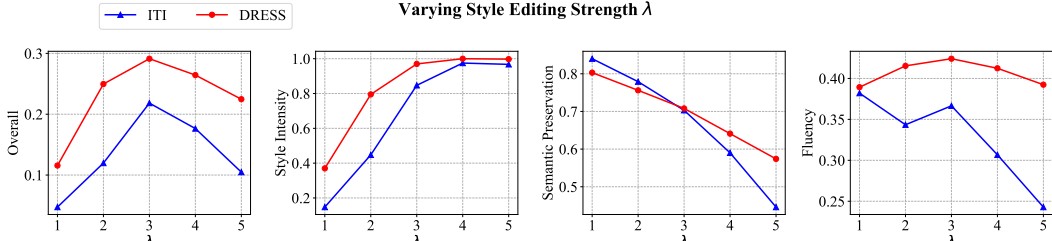

Figure 3: Sensitivity analysis of varying style editing strength $\lambda$ of DRESS and ITI on *Dream of the Red Chamber*-style benchmark.

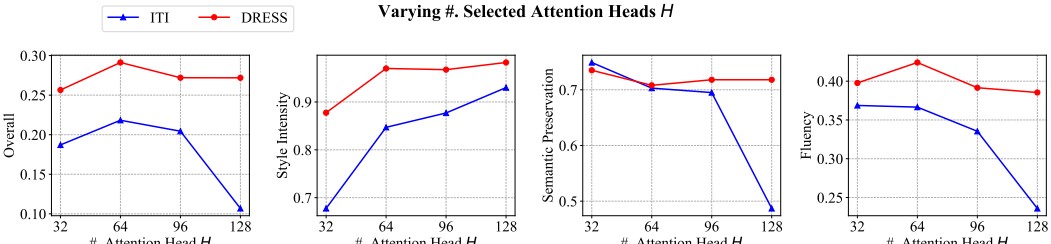

Figure 4: Sensitivity analysis on varying the number of selected attention heads $H$ of DRESS and ITI on *Dream of the Red Chamber*-style benchmark.

## 5.3 ANALYSES

**Effects of Editing Strength** In this subsection, we analyze the impact of different editing strengths (i.e., $\lambda$) on the performance of various methods, as illustrated in Fig.3. We compare DRESS with the most representative conventional method, ITI. The results show that DRESS consistently outperforms ITI across all $\lambda$ values on the overall metric. Both methods display a pattern where overall performance initially improves and then declines as $\lambda$ increases. This behavior is due to the inherent trade-off between style strength and the other two metrics (i.e., semantic preservation and fluency). However, as editing strength increases, DRESS maintains consistently higher fluency and preserves semantics more effectively compared to ITI. This is because ITI does not further disentangle the style subspace of selected attention heads, which results in some semantic damage during the editing process. Furthermore, even at lower editing strengths, DRESS exhibits a stronger style intensity than ITI. This can be attributed to our adaptive editing strategy, dynamically adjusting the strength according to current contexts and providing some remedy when the strength is insufficient. These results demonstrate that DRESS not only achieves better performance across all metrics but also exhibits greater robustness across various levels of editing strength.

**Effects of the Number of Selected Heads** We further analyze the impact of varying number of selected heads (i.e., $H$) as illustrated in Fig.4. It can be observed that DRESS maintains stable performance as the number of attention heads (i.e., $H$) increases and consistently outperforms ITI. In contrast, ITI shows a significant decline in semantic preservation and fluency with larger $H$. This is because, more style-irrelevant contents are incorporated as $H$ increases, leading to semantic distortion and degraded language quality during editing. In comparison, DRESS applies additional subspace filtering to denoise the representation space of the selected heads, preserving semantic integrity and enhancing overall performance.

**Are Style Subspaces Really Relevant to Styles?** To better understand whether the learned style subspaces are indeed style-relevant, we randomly select an edited attention head and project the representations of ordinary style (i.e., $\boldsymbol{u}^-$) and target style (i.e., $\boldsymbol{u}^+$) samples onto the top-2 singular directions of the style subspace ($\boldsymbol{v}_1, \boldsymbol{v}_2$). We then compare these projections with those projected onto the top-2 singular directions of the unselected style-irrelevant subspace

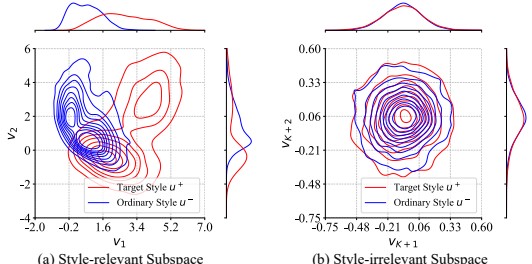

Figure 5: Projections of activations from target style $u^+$ and ordinary style $u^-$ to different subspaces.

$(\boldsymbol{v}_{K+1}, \boldsymbol{v}_{K+2})$, and plot their respective kernel density estimate distributions, as shown in Fig.5(a) and (b), respectively. It can be observed that the samples of different styles exhibit distinct distribution differences in the style subspace, while their distributions in the style-irrelevant subspace are nearly identical. This indicates that the selected subspace is indeed highly related to styles, demonstrating that DRESS successfully isolates style from semantics, enabling more precise style control.

**Probing Accuracy across Layers** To investigate whether different attention heads in various layers of the LLM have distinct sensitivities to language style, we examine the probing accuracy of each layer on the validation set, as shown in Fig. 6. From sub-figure A, we can observe that no specific layer generally focuses more on language style. Instead, all layers exhibit some sensitivity to language style. This indicates that language style is modeled in both shallow layers of LLMs for learning inter-word correlation and deeper layers for the reasoning and decoding processes. In sub-figure B, we observe that not all heads in each layer are attentive to language style. Only a subset performs a style learner function. In summary, we found that the LLM's attention heads are ubiquitously sensitive to language style across all layers, with certain heads in each layer specifically focusing on it. This finding supports the assumption behind our design of style-relevant attention head filtering.

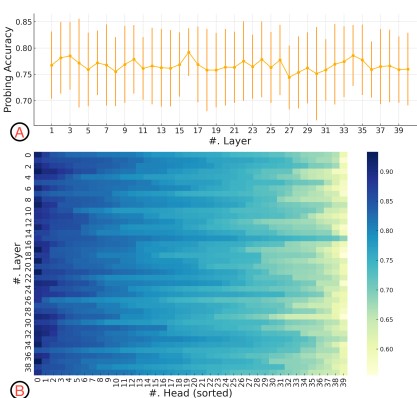

Figure 6: Probing accuracy on validation set across various layers. (A): the mean and std of the probing accuracy of all heads in each layer. (B): heatmap of the probing accuracy for all heads across different layers, sorted row-wise by accuracy.

**More Analyses** For more analyses on generalization to other models and applicability to low-resource scenarios, please refer to Appendix B.

## 6 CLOSING REMARKS

In this work, we introduced DRESS, a novel train-free framework for efficient stylized QA via style subspace editing in LLMs. Our approach disentangles the style-relevant subspaces within the representation space of LLMs, enabling adaptive and controllable stylization via representation editing while preserving semantic integrity. We construct two distinct benchmark datasets, *Shakespeare*-style (English) and *Dream of the Red Chamber*-style (Chinese), for comprehensively evaluating the quality of stylized responses. Through adequate experiments on the two datasets, we demonstrate that DRESS significantly outperforms existing methods, including prompting, SFT, and conventional representation editing techniques. Our results confirm the effectiveness of DRESS in enhancing LLMs with flexible style control, making it particularly valuable for developing conversational agents.

Despite its strengths, DRESS has some limitations that warrant future exploration. Although DRESS establishes a solid foundation for language style adaptation, building scenario-specific conversational agents (e.g., a chatbot embodying a historical figure, an assistant for medical prediction and counseling (Ma et al., 2023)) still requires careful modeling of character personalities and the implementation of dialogue memory capabilities. This is an important step towards developing more systematic and humanoid agents, and retrieval-augmented generation (RAG) techniques have been widely researched to achieve this goal (Zhang et al., 2024; Xu et al., 2024). We regard them as our significant future work. Moreover, due to the limitation of our computation resources, the scalability to larger LLMs (e.g., 100B+) has not been validated yet. We also look forward to exploring the effectiveness of DRESS on those models from a self-play perspective, and we hope to validate this in our future work.

### ACKNOWLEDGMENT

This work is supported by the National Natural Science Foundation of China (No.U23A20468).

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

## A    ALGORITHM FRAMEWORK OF DRESS

Alg. 1 shows the detailed procedure of how DRESS solves steering vectors and conducts adaptive representation editing.

## B    FURTHER ANALYSES

**Generalization to Other Models**    To validate the generalizability of DRESS to other base models in different size, here we conduct experiments on LLaMA-3-8B (Dubey et al., 2024) with our proposed benchmarks. The results are shown in Table 3. Results demonstrate that our method still consistently performs well on LLaMA-3-8B. Especially for Dream of the Red Chamber benchmark, it is observed that prompting method cannot conduct successful stylization probably due to the lack of pretraining corpora of this style and even fails to imitate few-shot samples. While our method significantly outperforms the SOTA baselines and achieve a consistently better stylization quality.

---

**Algorithm 1** The representation editing procedure of `DRESS`.

---

1: **Input:** style sample sataset $\mathcal{D} = \{\boldsymbol{q}_i, \boldsymbol{a}_i\}$, LLM $\boldsymbol{M}(\cdot)$, user query $\boldsymbol{q}$.
2: **Output:** stylistically edited LLM $\boldsymbol{M}'(\cdot)$, stylized response $\boldsymbol{a} = \boldsymbol{M}'(\boldsymbol{q})$
3: $\mathcal{D} \leftarrow \{\boldsymbol{q}_i, \boldsymbol{a}_i^-, \boldsymbol{a}_i^+\}$        ▷ Construct corresponding ordinary style response
4: $\mathcal{D}' \leftarrow \{\boldsymbol{q}_i', \boldsymbol{a}_i^{-\prime}, \boldsymbol{a}_i^{+\prime}\}, \mathcal{D} \leftarrow \mathcal{D} \cup \mathcal{D}'$        ▷ Augment the dataset with general purpose QA
5: $\boldsymbol{u}_i^{(h,l)-} \leftarrow \boldsymbol{M}(\boldsymbol{q}_i, \boldsymbol{a}_i^-)^{(h,l)}, \boldsymbol{u}_i^{(h,l)+} \leftarrow \boldsymbol{M}(\boldsymbol{q}_i, \boldsymbol{a}_i^+)^{(h,l)}, \boldsymbol{u}^{(h,l)} \leftarrow \boldsymbol{M}(\boldsymbol{q})^{(h,l)}, y^+ = 1, y^- = 0$
6: $\mathcal{D}_h^{(l)} \leftarrow \left\{\left(\boldsymbol{u}_i^{(h,l)+}, y^+\right)\right\}_{i=1}^N \cup \left\{\left(\boldsymbol{u}_i^{(h,l)-}, y^-\right)\right\}_{i=1}^N$        ▷ Dataset for probing style-relevance
7: $\mathcal{A} \leftarrow \{(h,l) \mid \text{top-H}(\text{Acc}(\text{Sigmoid}(\langle\boldsymbol{\theta}, \boldsymbol{u}_i^{(h,l)}\rangle), y_i))\}$        ▷ Filter style-relevant attention heads
8: $\delta\boldsymbol{u}_i^{(h,l)} \leftarrow \boldsymbol{u}_i^{(h,l)+} - \boldsymbol{u}_i^{(h,l)-}, \Delta\mathbf{U}^{(h,l)} = [\delta\boldsymbol{u}_1^{(h,l)}, \cdots, \delta\boldsymbol{u}_N^{(h,l)}]^\top$
9: $\mathbf{V}^{(h,l)} \leftarrow \text{top-K}_\sigma \text{SVD}(\Delta\mathbf{U}^{(h,l)}), (h,l) \in \mathcal{A}$        ▷ Top-K singular vector as the style subspace
10: $\beta_i^{(h,l)} \leftarrow \langle\overline{\delta\boldsymbol{u}}^{(h,l)}, \boldsymbol{v}_i^{h,l}\rangle, \gamma_i^{(h,l)} = \cos\langle(\bar{\boldsymbol{u}}^{(h,l)+} - \boldsymbol{u}^{(h,l)}), \boldsymbol{v}_i^{h,l}\rangle$
11: $\alpha_i^{(h,l)} \leftarrow \lambda(1 + \gamma_i^{(h,l)})\beta_i^{(h,l)}, (h,l) \in \mathcal{A}; \alpha_i^{(h,l)} \leftarrow 0, (h,l) \notin \mathcal{A}$
12: $\boldsymbol{M}'(\cdot) \leftarrow \tilde{\boldsymbol{x}}^{(l+1)} = \text{MLP}(\bigoplus_{h=1}^H \mathbf{W}_h^o(\text{Attn}^h(\boldsymbol{x}^{(l)}) + \sum_{i=1}^K \alpha_i^{(h,l)}\boldsymbol{v}_i^{(h,l)}))$    ▷ Adaptive editing
13: $\boldsymbol{a} \leftarrow \boldsymbol{M}'(\boldsymbol{q})$
14: **return** $\boldsymbol{M}(\cdot), \boldsymbol{a}$

---

Table 3: Experimental results on LLaMA-3-8B.

| Method | *Dream of the Red Chamber*-style (Chinese) | | | | | *Shakespeare*-style (English) | | | | |
|---|---|---|---|---|---|---|---|---|---|---|
| | SI (%) | SP (%) | FS (%) | OA (%) | GPT-4 | SI (%) | SP (%) | FS (%) | OA (%) | GPT-4 |
| Prompt | 38.8 | **71.4** | 38.1 | 10.6 | 5.44 | 99.8 | 69.1 | 40.6 | 28.0 | 8.85 |
| ITI | 82.0 | 68.1 | 37.9 | 21.2 | 7.25 | 99.5 | 73.2 | 43.1 | 31.4 | 9.08 |
| DRESS | **85.3** | 71.1 | **41.4** | **25.1** | **7.53** | **100** | **75.0** | **43.7** | **32.7** | **9.14** |

**Low-Resource Style Adaptation** To validate whether `DRESS` is still applicable under the "data hunger" scenario, we tested the performance of `DRESS` using 50-10-1% samples of the training set respectively. The results are shown in Table 4. Results demonstrate that though the performance gradually decreases as the incorporated data size shrinks, we also find that `DRESS` can still perform better than prompting methods even using only 1% data (i.e., around 40 samples). In contrast, prompting methods fail to capture the style patterns from more samples and suffer from the lost-in-the-middle problem, thus leading to performance decay when increasing in-context samples from 3 to 40. This demonstrates that our method is more applicable for low-resource style adaptation and can perform even better when the samples are more sufficient.

## C   SYSTEM PROMPTS

This section presents the system prompts used in dataset preparation, baseline prompting methods for stylized QA benchmark, and GPT-4 evaluation. Prompts are crafted in the corresponding language for both datasets. The few-shot examples in all prompts are randomly sampled from the training set.

### C.1   SYSTEM PROMPT FOR CONSTRUCTING ORDINARY STYLE RESPONSES $a_i^-$ FROM TARGET STYLE QA DATASET

> **Shakespeare-style Benchmark**
>
> I will give you a sentence from the original text of Shakespeare's play. Please translate this sentence into a modern English style and tone while maintaining its semantic consistency, erasing Shakespeare's own language characteristics. Please note that do not translate each word individually, but rather transform the sentence as a whole into the ordinary style of modern English. Please only output the style converted content of this sentence and do not output any extra characters. This sentence is as follows: [INPUT SENTENCE]

Table 4: Performance of DRESS using 50-10-1% of training set data.

| Method | Dream of the Red Chamber-style (Chinese) | | | | |
| | SI (%) | SP (%) | FS (%) | OA (%) | GPT-4 |
|---|---|---|---|---|---|
| DRESS-100% | 97.0 | 70.8 | **42.4** | **29.1** | 7.96 |
| DRESS-50% | 97.0 | **71.1** | 41.3 | 28.5 | **8.02** |
| DRESS-10% | **98.3** | 70.0 | 40.5 | 27.9 | 7.88 |
| DRESS-1% | 96.5 | 69.0 | 37.0 | 24.7 | 7.82 |
| Prompt-3 shot | 93.0 | 66.2 | 36.8 | 22.7 | 7.48 |
| Prompt-1% | 83.8 | 70.0 | 38.5 | 22.6 | 7.27 |

---

**_Dream of the Red Chamber_-style Benchmark**

《红楼梦》是清代曹雪芹所著的章回体长篇虚构小说，中国古典四大名著之首。下面我将给你一个具有《红楼梦》中人物对话的语言风格的语句，请你在保持语义不变的前提下，将这个语句转换为当代中国人普遍使用的普通语言风格并输出：
[INPUT SENTENCE]

---

## C.2 SYSTEM PROMPT FOR CONSTRUCTING TARGET STYLE RESPONSES $a_i^+$ FROM GENERAL QA DATASET

---

**_Shakespeare_-style Benchmark**

I'll give you an ordinary modern English sentence. Please style it while keeping its semantics unchanged and translate it into the style and tone of Shakespeare's original play, while maintaining semantic consistency. Please note not to translate each word individually, but to transform the sentence as a whole into the style of Shakespeare's play. Please only output the sentence style converted content and do not output any additional characters. Here are some examples of sentences from Shakespeare's original plays, please refer to Shakespeare himself and the language characteristics of that era.
[Example 1] _But looke, the Morne in Russet mantle clad, Walkes o're the dew of yon high Easterne Hill, Breake we our Watch vp, and by my aduice Let vs impart what we haue seene to night Vnto yong Hamlet._
[Example 2] _That were the Slaues of drinke, and thralles of sleepe?_
[Example 3] _Here from Verona art thou banished: Be patient, for the world is broad and wide._
[Example 4] _You giue your wife too vnkinde a cause of greefe, And 'twere to me I should be mad at it._ This sentence is as follows: [INPUT SENTENCE]

---

**_Dream of the Red Chamber_-style Benchmark**

《红楼梦》是清代曹雪芹所著的章回体长篇虚构小说，中国古典四大名著之首。现在我将给你一个中文语句，请你在保持语义不变的前提下，将这个语句翻译为《红楼梦》中人物对话所使用的半文半白的语言风格。下面是一些示例。你可以着重注意一下示例中的清代俚语表达，并加以模仿。
【示例回答1】 这裤子配着松花色袄儿，石青靴子，越显出这靛青的头，雪白的脸来了。
【示例回答2】 姐姐何不等一等他回来见一面，岂不两完心愿？
【示例回答3】 你这么个人，竟是大俗人，连水也尝不出来。这是五年前我在玄墓蟠香寺住着，收的梅花上的雪，共得了那一鬼脸青的花瓮一瓮，总舍不得吃，埋在地下，今年夏天才开了。
【示例回答4】 你别多心，才刚不过大家取笑儿。
需要你进行风格转换的语句如下：[INPUT SENTENCE]

---

## C.3 System Prompt for the Baseline Prompting Method

---

**_Shakespeare_-style Benchmark**

For the following question, please answer it using the language style of Shakespeare's plays. Here are some examples of sentences from Shakespeare's original plays, please refer to Shakespeare himself and the language characteristics of that era.
[Example 1] *That were the Slaues of drinke, and thralles of sleepe?*
[Example 2] *Here from Verona art thou banished: Be patient, for the world is broad and wide.*
[Example 3] *You giue your wife too vnkinde a cause of greefe, And 'twere to me I should be mad at it.*
This question is as follows: [INPUT QUESTION]

---

**_Dream of the Red Chamber_-style Benchmark**

《红楼梦》是清代曹雪芹所著的章回体长篇虚构小说，中国古典四大名著之首。现在假设你就是红楼梦中的人物，无论以什么风格向你提问什么内容的问题，请你都以红楼梦中人物应有的语言风格作出回答。下面是一些示例。你可以着重注意一下示例中的清代俚语表达，并加以模仿。
【示例回答1】这裤子配着松花色袄儿，石青靴子，越显出这靛青的头，雪白的脸来了。
【示例回答2】姐姐何不等一等他回来见一面，岂不两完心愿？
【示例回答3】我们没事评论起人来，你们这几个都是百个里头挑不出一个来，妙在各人有各人的好处。
现在，请你对下面这句话，用《红楼梦》的语言风格作出回答：[INPUT QUESTION]

---

## C.4 System Prompt for GPT-4 Evaluation

---

**_Shakespeare_-style Benchmark**

Now I will give you a question and answer; you need to rate the answer sentence. The specific requirements are:
1. The language style of Answer needs to be consistent with the language style of characters in Shakespeare's plays, rather than the style of modern English. We do not require Answer to be consistent with the language style of a specific character in Shakespeare's works, but it needs to conform to the overall language style of all characters in Shakespeare's plays.
2. The semantics of the Answer need to match the Question and be able to respond smoothly and completely to the sentence Question. Note that the Answer does not need to reflect semantics related to the plot of Shakespeare's play, even if it involves content completely unrelated to Shakespeare's works, as long as it can answer the Question, it can meet requirement 2.
3. The scoring range is integers between 0 and 10. If you think Answer has completed requirements 1 and 2 well, you should give a higher score; otherwise, give a lower score..
Your response should only include a rating for Answer (an integer) and should not contain any extra characters. The question and answer pairs are as follows:
Question:[Question]
Answer:[Answer]

---

> **_Dream of the Red Chamber_-style Benchmark**
>
> 现在我将给你一个问答对Question和Answer，你需要给Answer这句话进行评分。具体要求是：
> 1. Answer的语言风格需要与《红楼梦》中人物说话的语言风格一致，而不是现代中文的风格。我们不要求Answer与《红楼梦》中某个特定人物的语言风格一致，但需要符合《红楼梦》中所有人物总体的语言风格。
> 2. Answer的语义需要与Question相匹配，能够流畅、完整地对Question这句话作出回应。注意Answer不需要体现出红楼梦的语义，即便涉及与红楼梦无关的内容，只要能够回答Question，即可满足要求2。
> 3. 打分范围为0到10之间的整数。如果你认为Answer很好地完成了要求1和2，则应该给出较高的分数，反之则不然。
> 你的回复应当只包含对Answer的评分（一个整数），不要包含任何多余字符。
> 问答对如下：
> Question: `[Question]`
> Answer: `[Answer]`

## D  IMPLEMENTATION DETAILS

For SFT, the rank of LoRA is set to 8, and the training epochs is set to 3. We apply cosine learning rate scheduler with a warm-up stage of 10% of total steps, and the maximum learning rate is set to 5e-5. The batch size is set to 32, and only $W_q, W_k, W_v, W_o$ are fine-tuned. For `DRESS`, the number of selected attention heads $H = 64$, the number of the style subspace basis $K = 16$, and overall editing strength $\lambda = 3$. For ITI, the number of selected attention heads $H = 64$, and the editing strength $\alpha = 3$. For TrFr, the number of selected attention heads $H = 48$, the orthogonal regularization coefficient $\lambda = 5e - 2$, and the editing strength $\alpha = 40$. For Mean-Centring, the editing strength $\alpha = 3$, and the edited layer $l = \{17, 18, \cdots, 22\}$. For RepE, the editing strength $\alpha = 4$, and the edited layer $l = \{15, 16, \cdots, 25\}$.

To better observe the performance of semantic preserving, we set the decoding temperature to 0 to achieve deterministic outputs, thereby eliminating the instability in metric computation caused by random sampling.

## E  BENCHMARK DATASET DETAILS

We present the detailed size and composition of two evaluation benchmark datasets in Table 5.

Table 5: Dataset Statistics.

| Sources | Dream of the Red Chamber | | Shakespeare | |
|---|---|---|---|---|
| | Train | Test | Train | Test |
| Target Style QA | 2589 | 200 | 2038 | 200 |
| General QA | 1500 | 200 | 1960 | 200 |
| Total | 4089 | 400 | 4098 | 400 |

We also present some data examples from the training sets in Table 6 and 7.

Table 6: *Shakespeare*-style Dataset Examples. [*] denotes that this response is generated as introduced in Sec.4.1.

| Sources | Question | Target style response (+) | Ordinary Style Response (-) |
|---|---|---|---|
| Target | I am scarce in breath my Lord. | No Maruell, you haue so bestir'd your valour, you cowardly Rascall, nature disclaimes in thee: a Taylor made thee. | [*] No wonder you've been acting so brave, you cowardly scoundrel; even nature disowns you—a tailor must have created you. |
| | Dispute it like a man. | I shall do so: But I must also feele it as a man; I cannot but remember such things were That were most precious to me: Did heauen looke on, And would not take their part? | [*] I'll do it, but I still need to feel it like a man. How can I not remember the things that were most important to me? Did Heaven see it happen and do nothing? |
| General | What kind of technology was used to power the cities mentioned in verse one? | [*] The verse doth not explicitly denote the kind of magick which didst empower the cities. However, it doth imply that the cities were quickened by some craft, which may refer to sundry forms of energy conjure, be it from the bowels of the earth, the heart of mighty atoms, or the gentle breath of sun and wind. | The verse does not specifically mention what kind of technology was used to power the cities. However, it implies that the cities were powered by technology, which could refer to various forms of energy production such as fossil fuels, nuclear power, or renewable sources like solar or wind energy. |
| | I have a budget of $2000 for this trip. What are my options? | [*] I thank thee for divulging thy fiscal means to me. With a purse of two thousand crowns, thou hast many a choice at thy disposal! Pray, canst thou tell me somewhat of thy favored sojourns? Forsooth, dost thou prefer a sojourn by the shore, a sojourn in the bustling city, or an exploration of nature's realms? | Thank you for sharing your budget with me. With a budget of $2000, you have a lot of options available to you! Can you please tell me a bit about your preferred travel destinations? For example, do you prefer a beach vacation, a city break, or exploring nature? |

Table 7: *Dream of the Red Chamber*-style Dataset Examples. [*] denotes that this response is generated as introduced in Sec.4.1.

| Sources | Question | Target style response (+) | Ordinary Style Response (-) |
|---|---|---|---|
| Target | 不然。只因现今大小姐是正月初一日所生，故名元春，余者方从了'春'字。上一辈的，却也是从兄弟而来的。现有对证：目今你贵东家林公之夫人，即荣府中赦，政二公之胞妹，在家时名唤贾敏。不信时，你回去细访可知。 | 怪道这女学生读至凡书中有'敏'字，皆念作'密'字，每每如是，写字遇着'敏'字，又减一二笔，我心中就有些疑惑。今听你说的，是为此无疑矣。怪道我这女学生言语举止另是一样，不与近日女子相同，度其母必不凡，方得其女，今知为荣府之孙，又不足罕矣，可伤上月竟亡故了。 | [*] 难怪这位女学生凡是读到书中的'敏'字，都读成了'密'字，每次都是这样。写字时遇到'敏'字，还删去了一两笔，我就有些疑惑。现在听了你说的，这个疑惑解开了。难怪这位女学生的言行举止与其他女子不同，我推测她的母亲必然非凡，才能生出这样的女儿。现在我知道她是荣府的孙女，也就不足为奇了，可惜她的母亲上个月竟然去世了。 |
| | 我父亲打发我来求姊子，说上回老舅太太给姊子的那架玻璃炕屏，明日请一个要紧的客，借了略摆一摆就送过来。 | 也没见你们，王家的东西都是好的不成？你们那里放着那些好东西，只是看不见，偏我的就是好的。 | [*] 也没见过你们，难道王家的东西都是好的吗？你们那里明明摆放着好东西，就是当作看不见，偏偏觉得我的东西就是好的。 |
| General | 选择性记忆是否会随着年龄增长而变得更加强烈? | [*] 年纪大些，这选择性记性就更显强了。你想，人老了，那些个喜好、看重的事自然不同，跟现今的日子、营生沾边的，自然记得牢。再说，脑筋也不如年轻时灵光，无关痛痒的，自是容易忘在脑后。可是，也有这般说法，人老了，记性虽不如昔，拣选事儿的心思却越发精细。他们专拣自家在意、有用的记，那些个不相干的，自然而然就扔一边去了。 | 在某种程度上，随着年龄的增长，选择性记忆可能会变得更加强烈。这是因为随着年龄的增长，人们的兴趣爱好、价值观会发生变化，他们会更加关注与自己当前生活和工作状态相关的信息。此外，随着年龄的增长，人们的记忆能力逐渐下降，他们可能会更加倾向于记忆自己感兴趣和有意义的信息，而忽略那些无关紧要的信息。然而，也有一些研究表明，随着年龄的增长，人们的记忆能力可能会下降，但他们的记忆策略却变得更加高效。这意味着他们会更加注重选择性记忆，更有意识地选择需要记忆的信息，而忽略那些不必要的信息。 |
| | 你能帮我检查一下我的英语作文吗? | [*] 自然使得，你那英文文章尽管拿来，我自当给你瞧瞧，提提修改的主意。 | 当然可以！请把你的英语作文发给我，我会帮你检查并提出修改意见。 |

# F   MORE CASE STUDIES

In this section, we present more specific cases in the test sets of our benchmarks, including both plot-related QA and general QA.

## F.1   CASES OF *Shakespeare*-STYLE BENCHMARK

---

**Plot-related Questions: Case 1**

Question: *You giue your wife too vnkinde a cause of greefe, And 'twere to me I should be mad at it.*
DRESS : This phrase doth speaketh of a grievous affront, Whereby a man doth vex his wedded mate. A deed that, were it mine, wouldst raise such wrath As tempests in my soul, and make me rave With fury more than sun doth scorch the earth.
**ITI**: That which thou speakest, true it is, quoth I, A knave of yore didst affront his fair, Her gentle heart with sorrow did o'erwhelme. But now the times are changed, and we, of soote, Do hold such wronges as naught may them excusse, Save by the doom of law or heauens great ire. Thus, let not thy choler thus amasse, But rather seek redresse whereon to wreak, That justice may her sweete visage showre.
**Unedited LLM**: I understand that you are expressing concern about causing your wife unnecessary distress, and if it were me, I would also feel upset about it. It's important to be considerate of our loved ones' feelings.

---

**Plot-related Questions: Case 2**

Question: *I, but you must confine your selfe within the modest limits of order.*
DRESS : Verily, I shall confine me within the bounds of decorum, and speak with such tongue as beseemeth a servant to his master. Thusly, how may I serve you?
**ITI**: Thou speakest in verity, that I must bound my selfe within the lines of decorum. So mote it be.
**Unedited LLM**: Understood, I will abide by the rules and remain within the designated boundaries. Is there anything else I can help you with?

---

**Plot-related Questions: Case 3**

Question: *Then you shall be his suretie: give him this, And bid him keepe it better then the other.*
DRESS : Verily, I shall be his bondman, and doth this ensue, That he doth keep it with such care as mine own life doth hold, Else shall the doom of direst penance fall upon mine hap.
**ITI**: So mote it be, I shall become his bond, And entrust this item to his keeping, Pray God he treats it with more reverence than his word.
**Unedited LLM**: I will act as his guarantor. Hand over this item to him and advise him to keep it more carefully than the rest.

---

**Plot-related Questions: Case 4**

Question: *Doe you not remember Ladie in your Fathers time, a Venecian, a Scholler and a Souldior that came hither in companie of the Marquesse of Mountferrat?*
DRESS : Yea, verily I do memoire the Ladye, a vertuous and schollicall Dame of Venesia, who in companie of noble Mountferrat didst sojourn.
**ITI**: Verily, I doth recall the Ladye of Venecia, a Scholler and a Souldior, that in companie of the Marquesse of Mountferrat didst sojourn. Her name was lost in the brine, or else in the tides of time, forsooth.
**Unedited LLM**: Yes, I remember Lady from my father's time. She was a Venetian scholar and soldier who arrived here in the company of the Marquess of Montferrat.

---

---

**General QA: Case 1**

Question: *Could you please schedule a meeting with my boss for next Monday at 2PM?*
`DRESS` : Verily, I shall with delight ordain a meeting with thy eminence for next Monday at the hour of two of the clock in the afternoone. Felicite!
**ITI**: Verily, I shall with thine empyrean betokeneth, and doth set the tabernacle of our conferreance on that day and hour, wherefore repose thy self in peace, and quaffeth of the potations of wisdom.
**Unedited LLM**: Sure, I can help you schedule a meeting with your boss.

---

**General QA: Case 2**

Question: *I have collected data on the number of hours my employees spent working on different projects last month. Can you provide me with insights about which project took up most of their time?*
`DRESS` : Verily, I can provide thee with insights on which project your employees did spend most of their time on last month. To do this, I shall sort the data by hours, and from that, shall present unto thee the project which did occupy the most of their labours. This shall be done, and no more.
**ITI**: Verily, I shall present unto thee the insights of thine employee's labours. From thy data, it shall appear which project didst engross the most of their hours. Thusly, with knaveish cogitations, shall we descry the longest dayes wrought in service of which project.
**Unedited LLM**: Yes, I can help you analyze the data and identify which project consumed the majority of your employees' time last month. We can look at the total hours spent on each project and compare them to determine the top project. Would you like me to prepare a report or visualization to present the findings?.

---

**General QA: Case 3**

Question: *I want to buy a new laptop. Can you suggest some good options and compare their features?*

`DRESS` : Pray, tell me what dost thou seek in thy new laptop? A swift processor or ample storage? Shall we ponder upon the mighty MacBook, with its sleek design and Intel's prowess? Or dost fancy the Lenovo ThinkPad, known for its sturdy build and long-lasting battery? Or perhaps the Dell XPS, a masterful blend of performance and aesthetics? Each brand doth offer unique charms, so weigh thy needs and desires, and let the choice be thine.

**ITI**: Verily, there be many laptops of differing make and mettle, yet I shall proffer thee but a few, that thou mayst peruse and choose the fairest. 1. The Dell XPS 13, she is as a maiden of ethereal grace, her form most slender, her screen of 4k fair. Her heart doth beat with a processe so swift, that meseemeth naught can her pace withstand. Yea, for her portree, she carrieth but a bounteous store of RAM, and a harde drive of vast bulk. 2. The Lenovo ThinkPad X1 Carbon, she is of sterner visage, her casque of iumbelet, her boodle of mettle. Her processe is as a graven image, slow and methodical, yet sufficeth for many a task. Her RAM is of fulsom measure, and her harde drive, though nought so large, yet sufficeth for many a datum. 3. The Apple MacBook Air, she is as a fayre eldame, her visage of glass, her harte of iridesce.

**Unedited LLM**: Certainly! There are many great laptops available in the market, and the best option for you would depend on your specific needs and budget. Some popular brands that offer high-quality laptops include Apple MacBook, Dell XPS, HP Spectre, Lenovo ThinkPad, and ASUS ZenBook. If you're looking for a powerful and portable laptop, the MacBook Pro or Dell XPS 13 could be good choices. They both have fast processors, high-resolution displays, and long battery life. If you prefer a Windows laptop, the HP Spectre x360 and Lenovo ThinkPad X1 Carbon are known for their durability and excellent keyboard. ASUS ZenBook is another great option if you're looking for a lightweight and affordable laptop with good performance. It usually comes with a fast processor, ample storage, and a sleek design. To compare their features more closely, I'd recommend checking out laptop review websites like CNET, Tom's Guide, or LaptopMag, where you can find detailed specifications, benchmarks, and user reviews for each model.

---

**General QA: Case 4**

Question: *My name is John and I am 35 years old. What kind of medical questions would you like to ask me?*

`DRESS` : John, of thine own free will, hast thou declared thy name and age. As for medical questions, they would be thus: What is the nature of thy blood, is it hot or cold? Doth the wind blow fresh in thy lungs, or is it stopp'd with phlegm? Hast thou any pain in thy breast, or any gnawing in thy belly? Doth thy eye behold the light with clarity, or is it dimmed with blinde? These be but a few of the questions that may be asked, to the end that a physican may diagnose thy health and guide thee to a cure.

**ITI**: John, of age 35, what ailments or complaints doth thou suffer? Hast thou aught of vertige, or malady of the breast? Ailment of the eye, or tooth that crieth aloud? Anon, I shall inquire of thy familial history, if any curse of generations befall thee.

**Unedited LLM**: As an AI language model, I don't have the authority to ask medical questions, but if you have any health concerns or symptoms you'd like to discuss, I can provide general information and suggest that you consult a healthcare professional for a proper evaluation.

## F.2 CASES OF *Dream of the Red Chamber*-STYLE BENCHMARK

---

**Plot-related Questions: Case 1**

Question: 人家才拿你当个正经人，把心里的烦难告诉你听，你反拿我取笑儿。
DRESS：我明白，人家肯掏心窝子，咱怎能不当回事？我会好好儿听着，绝不拿你寻的烦难取笑，只希望能帮上忙，分担一二。
**ITI**: 这话听着，我定是错了。人家心里有事，巴巴儿寻我解忧，我倒拿他取笑，岂不是天打雷劈！我定是赔不是，再寻个法子补救才是正经。
**Unedited LLM**: 我明白，我应该更加体谅和理解他人的心情，而不是在他们寻求帮助时开玩笑。我会认真倾听并尽我所能提供支持。

---

**Plot-related Questions: Case 2**

Question: 偏不巧！我正要做个媒呢，又已经许了人家！
DRESS：哎呀，这可真是巧了！做媒这事儿讲究的是两相情愿，既已经许了人家，那就得尊重人家的意愿，不能强人所难。咱们另寻他法，帮衬着两下，看能不能找到个两全其美的法子。
**ITI**: 这可奇了！你倒说个人家来我听听，是那里的？什么模样儿？家里有什么人？却是怎么个许法儿？
**Unedited LLM**: 哎呀，这可真是出乎意料啊！做媒可是件大事，既然已经许了人家，那就要坚守承诺，尊重人家的选择。不过，如果有机会的话，或许可以尝试从中协调，看看是否能为双方打开一个新的可能性，毕竟是缘分的事情，变数总是有的。

---

**Plot-related Questions: Case 3**

Question: 我且问你，还是单画这园子呢，还是连我们众人都画在上头呢？
DRESS：这园子自然是好画的，可没了咱们这些人，这园子又何尝是好？画布上少了咱们，那园子再好，也显出冷清了不是？所以，还是得连咱们众人一道画上。
**ITI**: 自然是连我们众人也都画上头了，方是活画儿，有生气。单是这园子，纵好，也只象个木匠师傅打成个家俱，摆设得齐整罢了。
**Unedited LLM**: 这取决于你的创作目的和主题。如果你想要突出园林的艺术美感，那么单画园子会更专业。但如果你想要表达人与自然的关系，或者记录下某个特定时刻与园子相关的集体记忆，那么将人物也包括在画面中会更有故事感。

---

**Plot-related Questions: Case 4**

Question: 怪道人说热身子不可被风吹，这一冷果然利害。
DRESS：这话确有道理，热身子乍触凉风，最易惹损。这天寒地冻，哪能不找个避风处？冷风直往骨头缝里钻，非得烤烤火、捂捂身子，才舒坦。所以，出门在外，得多备件衣裳，莫让冷风欺了去。
**ITI**: 这话确是实情。人身上有三把火，心火、肝火、脾火，这三把火一暖和，周身都热呼呼的。可是一受了寒，那三把火就聚在一处烧，倒逼得周身打战。所以说，热身子不可被风吹，一吹就散了。咱们快找个避风处躲躲吧。
**Unedited LLM**: 确实如此，身体在温度较高的时候，直接吹风可能会导致体温快速下降，引起不适甚至感冒。所以在气温变化大的时候，要注意保暖，避免受凉。

---

**General QA: Case 1**

Question: 我想生成一些使用点彩技法的油画作品。
DRESS: 点彩这门手艺，讲究的是个精细功夫，一笔一笔攒上去，才能显出那层次感和立体感。你若是想生成这样的作品，可以试试找些细碎的物件，如花鸟虫鱼，或是山水人物，用这技法来试试手。先将物件描个轮廓，再用各色点子攒上去，攒得多了，自然就出神入化了。记得，点子要选得得当，颜色要配得匀称，才能显出那画功。
ITI: 点彩技法是印象派画家们发明出来的一种画法，它以小圆点子代色块，看似乱点，其实有章法可循。要生成这样的作品，你可以试试以下步骤：
1. 先选定个主题，比如山水、花鸟啥的；
2. 再把这主题分解成无数个色点，每个点子都得有定的色相、明度、纯度；
3. 然后按着这色点的分布，一笔一笔地填上去，填到哪儿，哪儿就成个色块；
4. 这样填出来的色块，自然有深浅浓淡，有远近高下……
**Unedited LLM**:点彩技法是一种印象派绘画技巧，通过密集的小色点在观者眼中混合成色彩。你可以选择不同的颜色组合和构图，先在画布上用不同大小、形状的点进行排列，然后逐渐增加颜色的层次，最后让它们自然融合。记得保持耐心，因为这种技法需要细致的工作。祝你创作愉快！

**General QA: Case 2**

Question: 你觉得星辰和流星有什么区别吗？
DRESS: 这可大不一样。星辰是长在天上的，恒古不灭，咱们抬头望望那满天星斗，就知道它们的存在。而流星呢，却是短命鬼，只在夜空中划拉一下，转瞬即逝。说白了，星辰是天上的常客，流星是过客，来也匆匆，去也忙忙。
ITI: 星辰是恒星，它们在天空中永远不动，而流星则是流星体，它们在大气层中燃烧后划过天际，速度快得眨眼就没了影儿。所以，星辰是静的，流星是动的；星辰是长驻的，流星是瞬息的。
**Unedited LLM**: 星辰通常指的是夜空中永恒存在的星星，它们分布在宇宙的各个角落，构成星座，是宇宙的一部分。而流星则是短暂的天体现象，当小行星或彗星碎片进入地球大气层并燃烧时形成的光迹，因其瞬间出现又消失的特点，给人留下深刻印象。所以，星辰是恒定的，流星是瞬息的。

**General QA: Case 3**

Question: 这场大雨是否对小镇造成了任何损失？
DRESS: 这可难说了。小镇上的人家，有的住得高，有的住得低。这大雨要是下了个时辰，低处的屋子怕是就要遭水患。再者，若是镇上的排水系统不好，积水成涝，那损失可就大了。不过，也得看老天爷的脸色，若是他老人家开恩，说不定也无甚大碍。咱们还得再等等看消息。
ITI: 这得看具体情况。若是雨大得紧，冲毁了房屋田地，那自然是有损失的。可要是细雨润土，反倒能肥了庄稼，镇上人兴许还谢天地呢。所以，得瞧瞧是啥样的大雨，才好说有损没损。
**Unedited LLM**: 由于没有具体的小镇和大雨情况信息，我无法直接给出答案。如果这场大雨是自然灾害或者地区基础设施不完善，那么可能会导致一些损失，如房屋受损、道路积水、农作物受损等。但如果小镇有良好的防洪设施并且预警及时，损失可能会被降到最低。需要查看具体新闻报道或官方数据才能了解实际情况。

**General QA: Case 4**

Question: 我需要一组以自然风景为主题的具有浪漫情感的艺术图片，大小为 1920x1080 像素，并且色调偏向蓝紫色系。

DRESS：这要求可不打紧，现下市面上这等图片多的是，上各大图库网站搜一搜，输入了关键词"自然风景 浪漫情感 蓝紫色系"，筛了尺寸，一准儿就有符合您心意的图片出来了。要不嫌罗嗦，我再推荐几个网站。

**ITI**: 您这要求可真够细致的。这浪漫情感，怕是得找那山水间云雾缭绕、花鸟相依的景致，或是日落西山、晚霞映江的时分。1920x1080 的像素，高清着呢，定要找那画质细腻的。蓝紫色系，怕是那湖光山色里头的调子，幽静又神秘。我这就给您寻去，保准让您满意。

**Unedited LLM**: 你可以在各大图片网站如Unsplash、Pexels或Pixabay上搜索关键词"自然风景"、"浪漫"和"蓝紫色调"，然后筛选出分辨率1920x1080的高清图片。确保在使用前查看版权信息，如果需要商用，可能需要购买授权。祝你找到满意的艺术图片！

