# OpenReview forum: "DRESSing Up LLM: Efficient Stylized Question-Answering via Style Subspace Editing"
_ICLR.cc/2025/Conference — ICLR 2025 Poster_

### Official Review · Reviewer_mVfE · 2024-10-31

**Soundness:** 3
**Presentation:** 3
**Contribution:** 2
**Rating:** 5
**Confidence:** 3

**Summary:**

This paper introduces an approach for style transfer. The main motivation is to learn a style-relevant subspace, and then project the common language model space to this space to enable stylized response. This work is similar in spirit to different model editing papers for knowledge editing. The only difference is that here the style is more or less an implicit form of knowledge. The paper suggests that the model is lightweight, training-free in terms of the inference.

**Strengths:**

1. The paper is well written and clearly motivated.
2. The paper's approach is technically sound.
3. The evaluation seems to show improvement.

**Weaknesses:**

1. The approach seems quite ad-hoc. Though it's not a prompt-based method, but it's trying to distill the prompt-based method into a low-rank representation space. From this perspective, it's still a prompt-based approach.
2. The prompt-based approach has many mentioned limitations. Also, it's really hard to make sure the content is exactly the same while only the style gets shifted. This error in the data synthesis pipeline will influence the performance of the method.
3. The experiments of this paper is highly limited to two styles. These two styles can be easily achievable by prompting. The paper was motivated to go beyond this, however, the experiments do not support that.

**Questions:**

The notation in section 4 is quite confusing.

1. I don't see the definition of N in equation (4). The lowercased n seems to indicate the dataset size. But the capital N seems to indicate a different thing?
2. the q_i in SVD shares the same notation as equation (3). But I think they are totally different things.
3. why is q missing in equation (5)? There isn't enough justification for that.

---

> ### Author Response · Authors · 2024-11-23
> **Rebuttal to Reviewer mVfE (1)**
>
> We sincerely appreciate your comments, suggestions, and every effort spent on reviewing our work. Here we attempt to address all your remaining concerns. In the following, we quote your comments and then give our detailed response point-by-point.
>
> > **W1. Is this a prompt-based method?**
>
> We would like to claim that the core idea and contribution of DRESS is **NOT simply prompt-based**. Firstly, our use of prompts is solely for constructing augmented data. As stated in the introduction, prompt-based methods often suffer from being **overly simplistic** (i.e., challenging to use plain text to describe a complex style), introducing **limited sample diversity** (i.e., only few-shot samples can be incorporated), thus producing outputs that are **unstable, incomplete, and overly rigid**. Therefore, we aim to leverage synthetic data generated by prompts to distill style-relevant representations **depicted by a larger volume of data and inject them into LLMs in a more flexible and robust manner**. The distillation procedure is **non-trivial** and makes our approach **fundamentally different from prompt-based methods**.
>
> Moreover, consider the increasingly popular paradigm of using closed-source LLM prompted synthetic data for tuning to improve LLM performance [1]. Could we identify this as a prompt-based method? We argue that methods **using prompts to construct data are not inherently "prompt-based."** The key contribution of those approaches, including ours, lies in **how we extract valuable information from synthetic data and inject it into the model effectively**. This is very different from a simple prompting approach.
>
> Finally, prompt-based methods may lack generalizability at the model level — a prompt effective for Model A might not work for Model B. In contrast, our approach is applicable to different models. Here we supplement experiments on LLaMA-3-8B model, and DRESS shows consistently significant performance improvements compared with prompt methods. (See Appendix B of our new version of paper or our rebuttal to Reviewer jiFa (2) for details.)
>
> **[Dream of the Red Chamber style]**
>
> | LLaMA-3-8B    | SI   | SP   | FS   | OA   | GPT4  |
> |---------------|------|------|------|------|-------|
> | 3-shot Prompt | 38.8 | 71.4 | 38.1 | 10.6 | 5.44  |
> | ITI           | 82.0 | 68.1 | 37.9 | 21.2 | 7.25  |
> | DRESS         | 85.3 | 71.1 | 41.4 | 25.1 | 7.53  |
>
> **[Shakespeare style]**
>
> | LLaMA-3-8B    | SI   | SP   | FS   | OA   | GPT4  |
> |---------------|------|------|------|------|-------|
> | 3-shot Prompt | 99.8 | 69.1 | 40.6 | 28.0 | 8.85  |
> | ITI           | 99.5 | 73.2 | 43.1 | 31.4 | 9.08  |
> | DRESS         | 100  | 75.0 | 43.7 | 32.7 | 9.14  |
>
> To summarize, our method is NOT a so-called “prompt-based approach”.
>
> [1] Toshniwal S, Moshkov I, Narenthiran S, et al. OpenMathInstruct-1: A 1.8 million math instruction tuning dataset[C]. NeurIPS 2024 Dataset & Benchmark Track (Oral).
>
> > **W2. The influence of errors in the data synthesis pipeline.**
>
> First of all, it is true that we cannot ensure that the augmented semantics are completely consistent with the original samples. Yet most of them are quite similar, and we can assume that **the deviation between the generated and original samples is slight and can be regarded as “noises”**. In Sec. 4.2 & 4.3, we highlight that our method incorporates a **denoising module** to mitigate the impact of those noisy errors. We **selected the attention heads that are most related to styles** to minimize the effects of semantics. We also use the **main component singular vectors of SVD to span the style subspace**, which allows us to eliminate the noises from those slight semantic deviations and **identify a more robust style subspace for editing**. Therefore, in spite of the imperfect errors during the data synthesis procedure, our approach further eliminates those influences and performs robustly with better style adaptation, more fluent expressions, and well-preserved semantics.

---

> ### Author Response · Authors · 2024-11-23
> **Rebuttal to Reviewer mVfE (2)**
>
> > **W3. Style adaptation on our benchmark is easily achievable through prompting?**
>
> In our work, we selected two representative language styles from the most widely used and the most natively spoken languages (i.e., English and Chinese). Both are from an earlier era, where there are much **fewer corpora of the same style that could be inherently learned in the pretrained LLMs**. Hence, those two benchmarks are very challenging since LLMs do **NOT** have sufficient inherent knowledge to imitate the style **solely with prompts**. Compared with other types of styles (e.g., emotions, tones such as humorous), LLMs can easily adapt based on the data they have encountered, which does **NOT** pose a significant challenge. Moreover, adapting to our selected styles requires adjustments in multiple aspects, including differences in vocabulary between early modern and contemporary language, sentence construction logic, rhetorical techniques, and whether expressions are implicit or explicit, etc. These adaptations demand a very high level of abstraction and attention to detail to achieve proper alignment. In conclusion, our constructed benchmark is **comprehensive, representative, generalizable and challenging**, which is one of the core contributions of this paper.
>
> The quantitative experimental results on our benchmarks show that DRESS **outperforms all other baselines**, including prompting methods, **by a significant margin**, especially on Dream of the Red Chamber dataset. Actually, as we mentioned in the response to W1 and Intro, prompting methods often suffer from the rigidity of plain-text description and the difficulties of incorporating sufficient in-context demonstrations. Moreover, prompts **cannot explicitly separate the style and semantics apart as we did**, which further influences the adaptation quality. The results and analyses clearly demonstrate that there is still significant room for improvement in style adaptation quality when relying solely on prompting on our benchmark. This underscores that our benchmark is **NOT** as easily achievable as commented.
>
> > **Q1/2. Questions about notations:**
>
> Sorry for all the typos we made in the equations, and we have fixed them in the new version of our manuscript. Specifically,
>
> 1.	We change the notation of $Q$ in Eq.(3) to $S$ to avoid the confliction with the notation of questions q.
>
> 2.	$N$ is actually the sum of the target style QA dataset size $n$ and general QA dataset size $n’$, i.e., $N=n+n’$. There is a lack of this explanation in our previous version and we have supplemented it in the new manuscript.
>
> > **Q3. Why not incorporate $S$ (originally $Q$) in Eq.(5)**
>
> In SVD ($\Delta U = S \Sigma V$), $S$ is used for depicting the eigenvectors of the **column space** of $\Delta U$, where most information is about the **covariance between samples**. While $V$ depicts the eigenvectors of the **row space, i.e., the hidden representation spaces**. After selecting top-K singular vectors of $V$, the physical meaning of $V$ is the orthogonal bases of the style subspace, while $S$ turns out to be the **“word embeddings”** of the $N$ samples under the selected $V$ bases – **depicting the coordinates of the samples** in the linear space spanned by $V$. Therefore, under this context, $S$ should not be used in depicting the style-relevant representation subspace.

---

> ### Author Response · Authors · 2024-11-30
> **A kind reminder as the deadline is approaching**
>
> Dear Reviewer mVfE,
>
> I hope this message finds you well.
>
> I am writing to kindly remind you that the deadline for the discussion phase regarding our paper is drawing close. We greatly appreciate the time and effort you have already dedicated to reviewing our work, and we have made point-by-point responses to your constructive suggestions and questions. However, it appears that we have not yet received your response to our rebuttal, and we are wondering if we have addressed your concerns so far. Your insights and feedback are invaluable to us, and we hope our explanations could make it clearer for you to understand our contributions.
>
> After carefully addressing the comments and corresponding revisions, we kindly request your consideration of a more positive evaluation of our work if deemed appropriate. If you require any additional information or clarification from our end, please do not hesitate to reach out to us. We are more than willing to assist you in any way possible to ensure a thorough and constructive review process.
>
> Thank you once again for your time and consideration. We are looking forward to hearing from you soon.
>
> Warm Regards and Happy Thanksgiving!

---

> > ### Comment · Reviewer_mVfE · 2024-11-30
> > **Reviewer feedback**
> >
> > Dear authors,
> >
> > Thanks for providing detailed response to my review. The SVD confusion was well addressed. Thank you.
> >
> > For the prompt-based method, I didn't mean to say that your method is "prompt-based". I initially meant that your method is capped by the prompt-based approach from GPT-4. Then you distill this dataset back to the editing vector. The datasets being evaluated are rather niche. The improvement over "prompt-based model" is over "Qwen", rather than over "GPT-4". It's not fair comparison given that your Qwen model is tuned with the style transfer. You should actually compare with GPT-4 based prompting method.
> >
> > With the previous reasons, I like to retain my score.

---

> ### Author Response · Authors · 2024-12-01
> **Further Explanations to Reviewer mVfE**
>
> Thank you for your response. However, I think there might still be some misunderstanding of our work.
>
> Firstly, our method is **model-agnostic**, meaning that DRESS should be discussed and compared with other algorithms **independently of the model selection**. From the perspective of experimental principles, it would be unfair to compare algorithm A equipped with GPT-4 to another algorithm B equipped with Qwen to demonstrate the effectiveness of the algorithms, as their results are heavily influenced by the models’ capabilities. **Our focus is on the effectiveness of different editing methods, NOT on model benchmarking; and our primary contribution is a generalizable editing strategy, NOT an edited model product**. (The generalizability is tested on LLaMA-3-8B in Appendix B.) In other words, for a fair evaluation, if a direct comparison between DRESS and "prompting methods of GPT-4" is required, it would necessitate testing DRESS on GPT-4 as well. Unfortunately, this is impractical as **GPT-4 is a closed-source model**, which prevents access to its representation space.
>
> Secondly, we’d also like to claim that **the performance of our method will NOT be capped/upper bounded by GPT-4**, as **DRESS could theoretically be applied to GPT-4 for further improvements if we can obtain GPT-4 representation space**. In fact, there are unaligned semantic noises and imperfect style adaptation cases in current GPT-4 generated data. E.g.,
> ```
> Query: "你认为在团结合作中最重要的是什么？",
> GPT-4 Generated Stylized Response: "团结协作，首要的便是互信互助。唯有互信方能结成牢固的联盟，共谋大业。至于尊重，则是维系团结的基石，尊重他人的见解、功绩，方能营造出和睦的团队气氛，激发出众人共创的智慧与热忱。如此看来，团结合作的真谛，便是那信任与尊重的交织。"  ## Insufficient stylization, still looks like contemporary modern Chinese
> Original Response : "在团结合作中，最重要的是相互信任和相互尊重。只有相互信任，才能建立起紧密的合作关系，共同实现目标。而相互尊重则是保障团结合作的前提，只有尊重彼此的想法、意见和贡献，才能建立起和谐的团队氛围，激发每个人的创造力和积极性。所以说，团结合作的关键是相互信任和相互尊重。"
> ```
> ```
> "Query": "我需要翻译这句日语：“おはようございます”。",
> GPT-4 Generated Stylized Response: "这是日本话，讲的是“早安”之意。咱们大清早的，这会儿也该起来了。" ## Irrelevant contents incorporated, unaligned semantics with the original response.
> Original Response: "这句日语的意思是「早上好」。"
> ```
> Whereas DRESS applies attention head filtering, style subspace filtering, and adaptive editing to alleviate those noises and learn a more proper style adaptation strategy. **This process is also applicable to GPT-4 to further enhance itself.** In fact, This approach represents a **“self-play” mechanism**, where models improve themselves from the process of **correcting and rethinking their self-generated data**. Though we are currently unable to apply DRESS directly to GPT-4 due to its closed-source nature (as discussed in our limitations, Section 6), we try to show that **DRESS can also be used as an effective self-play technique**. Specifically, we conduct experiments on Qwen-1.5-14B using Qwen-1.5-14B itself as the model for data synthesis and observe the performance as below.
>
> | Qwen-1.5-14B                        | SI   | SP   | FS   | OA   | GPT4  |
> |-------------------------------------|------|------|------|------|-------|
> | Prompt                              | 93.0 | 66.2 | 36.8 | 22.7 | 7.48  |
> | DRESS (GPT4 generated data)         | 97.0 | 70.8 | 42.4 | 29.1 | 7.96  |
> | DRESS (Qwen-1.5-14B generated data) | 98.0 | 70.1 | 40.2 | 27.6 | 7.88  |
>
> It can be demonstrated that **DRESS+Qwen augmented data could outperform prompt+Qwen itself**. This shows the self-play potential of DRESS, incorporating more precise style representations, fewer noises, and more flexible style control to break the capability bound of the LLM where the data are augmented. **This result gives us the confidence to deduce that our method could also be able to enhance the style adaption ability of GPT-4**. Moreover, it is also true that data with higher quality can improve the effectiveness of our method (DRESS+GPT-4 data v.s. DRESS+Qwen data), also demonstrating the significance of distilling knowledge from stronger LLMs for improvements.
>
> Finally, we further emphasize that it is very meaningful to study how to improve smaller models. In tasks such as NPC creation, where hundreds of AI agents may be required, the computational cost of using large models like GPT-4 for each agent becomes prohibitive. Considering that the capability of small models is limited, efficient and effective style adaptation techniques like DRESS are crucial for enhancing their utility in such applications.
>
> We hope our explanations could resolve your problems. Thank you once again for your time and efforts in replying and considering our rebuttal.

---

> > ### Comment · Reviewer_mVfE · 2024-12-01
> > **Response**
> >
> > Thanks for the self-improvement experiments. The results do look more promising. The main bottleneck of this paper now lies in its evaluation and general applicability. The paper targets for a broad range: "In tasks such as NPC creation, where hundreds of AI agents may be required". However, the evaluated benchmarks are rather limited in styles. In my experience, persona is not only about style but also about personalized knowledge. The paper only tackles one part of this problem.
> >
> > Despite these concerns, I would like to raise my score to 5, which is slightly below acceptance. I still believe the paper could be enhanced by more general evaluation and more broader user studies.

---

> ### Author Response · Authors · 2024-12-02
> **Further Explanations to Reviewer mVfE (2)**
>
> Thank you for your prompt reply and proactive engagement. For the application scenario, our ultimate goal is indeed to build humanoid AI agents. However, it is a very huge roadmap, and stylization is the first step we are going to take. In Section 6, we have discussed our future works, including considering more detailed personality building and memory capabilities, etc. We are very pleased to find that your thoughts align with ours and we will keep on working on it. Back to the paper, we are actually focusing on this very first step to take toward our ultimate goal, and we present 1) convincing techniques, 2) two very representative and challenging benchmarks (we will keep updating more), to build a cornerstone for our roadmap. We think this is a fundamental step for the humanoid AI agent community. For the further steps, we believe they deserve another research paper to contain.
>
> Thank you once again for your comments and suggestions. We hope our explanations could resolve your problems, and we remain committed to addressing any remaining concerns and ensuring a better quality of our work.

---

### Official Review · Reviewer_AsRZ · 2024-11-01

**Soundness:** 3
**Presentation:** 3
**Contribution:** 3
**Rating:** 8
**Confidence:** 4

**Summary:**

Stylized generation to reference style is an important problem in language generation. This paper introduces a new approach for stylized generation with LLM that differs from prompting and fine-tuning based approaches. To ensure that the generation quality is not compromised in the pursuit of stylized generation, the paper introduces the notion of steering vectors that is learnt by filtering the appropriate attention head that controls the style aspects of the generation and filtering the irrelevant style subspaces to minimize the impact on semantics while maximizing the impact on style. To evaluate the approach, the paper further introduces 2 benchmarks: Shakespeare style English QA and Dream of Red Chamber styled Chinese QA.

**Strengths:**

The approach is interesting and builds on top of existing works. There is a lot of value to the solution and the extensive evaluation is impressive.

**Weaknesses:**

Given the problem is being tackled by several researchers over the past few years, I would have loved to see more details on the "data hunger" of the proposed approach. For e.g., a prompting based approach requires only a few samples and if the current approach is only marginally better than that in qualitative comparisons, the value might be ambiguous. To this end, I would like to see a bit more detailed comparison along the lines of the data needs - which is critical to extend to low-resource style adaptation cases.

**Questions:**

Seen the weakness section above.

---

> ### Author Response · Authors · 2024-11-23
> **Rebuttal to Reviewer AsRZ**
>
> We would like to express our genuine gratitude for your time and efforts on reviewing our work. We are pleased to hear that you find our work interesting and valuable.
>
> The “data hunger” issue you mentioned is indeed very crucial to validate the applicability of our method under a low-resource setting. Hence, we conduct experiments on both benchmarks using randomly sampled 50%-10%-1% of the entire training set and test their performance respectively. The results are shown below.
>
> | Dream of the Red Chamber | SI   | SP   | FS   | OA   | GPT4  |
> |--------------------------|------|------|------|------|-------|
> | DRESS                    | 97.0 | 70.8 | 42.4 | **29.1** | 7.96  |
> | DRESS-50%                | 97.0 | 71.1 | 41.3 | 28.5 | **8.02**  |
> | DRESS-10%                | 98.3 | 70.0 | 40.5 | 27.9 | 7.88  |
> | DRESS-1%                 | 96.5 | 69.0 | 37.0 | 24.7 | 7.82  |
> | Prompt-3 shot            | 93.0 | 66.2 | 36.8 | 22.7 | 7.48  |
> | Prompt-1%                | 83.8 | 70.0 | 38.5 | 22.6 | 7.27  |
>
> Results demonstrate that though the performance gradually decreases as the incorporated data size shrinks, we also find that DRESS can still perform better than prompting methods **even using only 1% data (i.e., around 40 samples)**. In contrast, prompting methods fail to capture the style patterns from more samples and suffer from the lost-in-the-middle problem, thus leading to performance decay when increasing in-context samples from 3 to 40. This demonstrates that our method is **more applicable for low-resource style adaptation and can perform even better when the samples are more sufficient**.
>
> Thank you again for your invaluable suggestions. We have supplemented those results in our paper (Appendix B).

---

> ### Author Response · Authors · 2024-11-30
> **A kind reminder as the deadline is approaching**
>
> Dear Reviewer AsRZ,
>
> I hope this message finds you well.
>
> I am writing to kindly remind you that the deadline for the discussion phase regarding our paper is drawing close. We greatly appreciate the time and effort you have already dedicated to reviewing our work, and we have made point-by-point responses to your constructive suggestions and questions. However, it appears that we have not yet received your response to our rebuttal, and we are wondering if we have addressed your concerns so far. Your insights and feedback are invaluable to us, and we believe they have significantly enhanced the quality of our manuscript.
>
> After carefully addressing the comments and corresponding revisions, we kindly request your consideration of a more positive evaluation of our work if deemed appropriate. If you require any additional information or clarification from our end, please do not hesitate to reach out to us. We are more than willing to assist you in any way possible to ensure a thorough and constructive review process.
>
> Thank you once again for your time and consideration. We are looking forward to hearing from you soon.
>
> Warm Regards and Happy Thanksgiving!

---

> > ### Comment · Reviewer_AsRZ · 2024-12-01
> >
> > Thanks to the authors for the response and new results, I lean towards accepting it.

---

> > > ### Author Response · Authors · 2024-12-01
> > > **Thank you**
> > >
> > > Thank you for your time, effort, and positive assessment. Your guidance and insights have been invaluable in refining our research. We remain committed to addressing any remaining concerns and ensuring a better quality of our work.

---

### Official Review · Reviewer_DDuX · 2024-11-04

**Soundness:** 3
**Presentation:** 3
**Contribution:** 3
**Rating:** 6
**Confidence:** 3

**Summary:**

The paper introduce DRESS, a new method for making LLMs generate answers in specific style without need training. Main contributions:
* Propose style subspace editing technique that find and modify specific parts of LLM representations to control output style
* Use 3 key techniques: attention head filtering, style subspace filtering, adaptive editing strength
* Create 2 benchmark datasets (Shakespeare-style English, Dream of Red Chamber-style Chinese) for testing stylized QA
* Show better results than previous methods like prompting, fine-tuning, etc.

**Strengths:**

* The proposed approach is novel, interesting and intuitive, instead of traditional prompting/fine-tuning, they find and edit "style subspace" in LLM representations. This is more efficient because no training needed
* The paper has good theoretical foundation, they use concepts like orthogonality of representations and attention head functions to justify their method
* Implementation details are clearly presented in the paper, they explain exact math formulas and algorithms, make it easy to reproduce
* This paper conducted comprehensive experiments, which test on 2 very different language styles (English + Chinese), compare with strong baselines, use multiple evaluation metrics. The results look convincing, both automatic metrics and GPT-4 ratings show clear improvements over baselines

**Weaknesses:**

* Need more analysis why method work better, authors show good results but don't explain deeply why style subspace editing is better than other approaches
* Some technical terms not explained well - like "style-relevant subspace", "adaptive editing strength" - need more intuitive explanation
* There are limited style types tested, only try 2 literary styles, should test more modern styles like formal/casual, different emotions, etc.
* Lacking of through discussion of limitations
* This paper's ablation studies are not enough - should test each component (attention filtering, subspace filtering, adaptive editing) separately to show importance

**Questions:**

Have you tried methods similar to knowledge editing?

---

> ### Author Response · Authors · 2024-11-23
> **Rebuttal to Reviewer DDuX (1)**
>
> We sincerely appreciate your comments, suggestions, and every effort spent on reviewing our work. Here we attempt to address all your remaining concerns. In the following, we quote your comments and then give our detailed response point-by-point.
>
> > **W1. Explanations on why style subspace editing is better:**
>
> In the Introduction and experimental results, we have analyzed our advantages over other methods. Here we specify it and provide some deeper explanations. As we mentioned in our paper, our method conducts better stylization for its **semantic-isolation property and flexible adaptive edition strategy**. During editing, we isolate the style representation space from semantics and edit only within the style subspace to ensure the semantic integrity and expression accuracy. The adaptive editing strengths also enable more flexible adjustment to the generation of every token, which enhance the fluency and make the stylization well-controlled.
>
> Delving deeper, our proposed method disentangles style-invariant representations for **better generalization to various queries** and shows consistently better performances. Previous works [1, 2] and surveys [3] also demonstrate that feature disentanglement techniques, which encourage the separation of domain-invariant (i.e., general knowledge and semantics here) and domain-specific (i.e., style) features, and has been **practically effective in controlling the invariance of the general domain-irrelevant knowledge and benefit generalization to different domains**. This also explains the effectiveness of our proposed methods.
>
> [1] Bengio Y, Courville A, Vincent P. Representation learning: A review and new perspectives[J]. IEEE TPAMI, 2013, 35(8): 1798-1828.
>
> [2] Ilse M, Tomczak J M, Louizos C, et al. Diva: Domain invariant variational autoencoders[C]. Medical Imaging with Deep Learning. PMLR, 2020: 322-348.
>
> [3] Wang J, Lan C, Liu C, et al. Generalizing to unseen domains: A survey on domain generalization[J]. IEEE TKDE, 2022, 35(8): 8052-8072.
>
> > **W2. More intuitive explanations of technical terms**
>
> We’d like to apologize for the inconvenience and below are some detailed explanations. “Style-relevant Subspaces” refers to the subspace within the original LLM representation space highly related to the language style but weakly related to semantics. (Also see the beginning of Sec.4.3 for the insights of proposing this concept.) “Adaptive editing strength” refers to the dynamic adjustments of how much edition of each aspect of the style (i.e., each basis of the style subspace) that the current generating token should be applied, as different aspects of a style may have varying importance or influence depending on the specific context. (Also see the beginning of Sec.4.4 for the insights of proposing this concept.)
>
> > **W3. The limitations of our selected style benchmarks**
>
> In our work, we selected two representative language styles from the most widely used and the most natively spoken languages (i.e., English and Chinese). Both are from an earlier era, where there are much **fewer corpora of the same style that could be inherently learned in the pretrained LLMs**. Hence, those two benchmarks are very **challenging** since LLMs do **NOT** have sufficient inherent knowledge to imitate the style solely with prompts. Compared with other types of styles (e.g., emotions, tones such as humorous) mentioned in the comments, LLMs can easily adapt based on the data they have encountered, which does not pose a significant challenge and cannot demonstrate the superiority of our designed methods.
>
> Moreover, adapting to those styles requires adjustments in multiple aspects, including differences in vocabulary between early modern and contemporary language, sentence construction logic, rhetorical techniques, and whether expressions are implicit or explicit, etc. It demands a very high level of abstraction and attention to detail to achieve proper adaptation. Last but not least, our benchmark is a more **realistic** experimental setting, as building an interactive conversational agent (e.g., NPCs, character AI, etc.) requires a **comprehensive imitation of multiple aspects of language styles instead of adapting to some specific aspects like tones**.
>
> In conclusion, our constructed benchmark is **comprehensive, representative and challenging**, which is one of the core contributions of this paper.

---

> ### Author Response · Authors · 2024-11-23
> **Rebuttal to Reviewer DDuX (2)**
>
> > **W4. Lacking of thorough limitations**
>
> In Paragraph 2 of Sec.6, we initially propose some limitations of our proposed techniques, i.e., the modeling of character personalities and the implementation of dialogue memory capabilities. Now we have supplemented some discussions on the scalability of our methods in the new version.
>
> > **W5. More ablation studies.**
>
> We have originally presented DRESS* as one ablation study, where the effectiveness of adaptive editing strategy is validated.
>
> For attention head filtering, **we have conducted sensitivity analysis on varying the number of selected attention heads** (See Sec.5.3 “Effects of the Number of Selected Heads” and Fig.4 for detailed conclusions). Here we supplement the ablation study on discarding the attention head filtering module (i.e., keeping all attention heads, H=1600) and the results are shown as follows.
>
> | Dream of the Red Chamber                | SI   | SP   | FS   | OA    |
> |-----------------------------------------|------|------|------|-------|
> | DRESS, H=32(default)                    | 97.0 | 70.8 | 42.4 | **29.1**  |
> | DRESS, H=128(also in Fig.4)             | 98.3 | 71.8 | 38.5 | 27.2  |
> | DRESS, H=1600(all attention heads kept) | 97.5 | 51.4 | 29.9 | 15.0  |
> | ITI, H=128                              | 93.0 | 48.7 | 23.6 | 10.7  |
>
> We can observe that **keeping all attention heads edited will lead to a great performance decay**, demonstrating the significance of conducting attention head filtering. However, **DRESS (H=1600) still performs much better than ITI (H=128) with around 12x more style-irrelevant heads incorporated**. This is because DRESS conducts further **denoising and adaptive edition** that relieve the problem of incorporating too many style-irrelevant components. This also demonstrates that effectiveness of our design.
>
> For the ablation study on style subspace filtering, as our **adaptive editing strategy is strongly bonded with the results of the solved style subspace** (i.e., the adaptive editing strength is attached to each basis in the style subspace). Hence, to validate the effectiveness of style subspace filtering, we should **apply the attention head filtering strategy only** for the ablation study. However, this setting **falls back to the ITI method**, and our results have shown significant improvements, demonstrating the effectiveness of style subspace filtering.
>
> > **Q1. Have we tried methods similar to knowledge editing?**
>
> To the best of our knowledge and according to survey [4], knowledge editing can be mainly categorized into two mechanisms: fine-tuning and prompting, which are **already included in our selected baseline methods**. The difference is that those methods are specifically re-designed for the knowledge editing tasks. However, there is a huge gap between knowledge editing and style adaptation. Knowledge editing aims to modify the pretrained knowledge and concepts within the LLM parameters, whereas style adaptation aims to keep them with only language style conversions. Hence, the methods used for knowledge editing are somewhat contradictory to the goal of style adaptation. Therefore, we use the modest fine-tuning and few-shot prompting methods as our comparison baselines.
>
> [4] Wang S, Zhu Y, Liu H, et al. Knowledge editing for large language models: A survey[J]. ACM Computing Surveys, 2023.

---

> > ### Comment · Reviewer_DDuX · 2024-11-27
> > **Feedback to Author Response**
> >
> > Thanks to the authors for the response and new ablation results, which have addressed my concerns about this paper. I lean towards accepting it.

---

> > > ### Author Response · Authors · 2024-11-27
> > > **Thank you**
> > >
> > > Thank you for your time, effort, and positive assessment. Your guidance and insights have been invaluable in refining our research. After carefully addressing the comments and revising the manuscript accordingly, we kindly request your consideration of a more positive evaluation of our work, if deemed appropriate. We remain committed to addressing any remaining concerns and ensuring a better quality of our work.

---

### Official Review · Reviewer_jiFa · 2024-11-05

**Soundness:** 2
**Presentation:** 3
**Contribution:** 3
**Rating:** 6
**Confidence:** 4

**Summary:**

This paper introduces DRESS, i.e., Disentangling Representation Editing in Style Subspace, a new approach for generating stylized responses from Large Language Models (LLMs) via representation editing. The core idea is to disentangle and edit within style-relevant subspaces of the LLM's representation space, enabling training-free style adaptation while preserving semantic meaning. Empirical results on the curated benchmark show that DRESS outperforms baselines including prompting, SFT, and other representation editing approaches.

**Strengths:**

- The proposed approach is well-motivated and intuitive. The approach has clear advantages over previous methods such as prompting and SFT.
- The training-free approach achieves superior performance compared to supervised fine-tuning, which shows the effectiveness of DRESS.
- The paper’s writing and presentation are clear and easy to follow.

**Weaknesses:**

- The dataset construction process heavily relies on GPT-4 to collect the stylized responses, which could be limited and biased with GPT-4’s capability.
- The evaluation benchmark only considers two language styles, one for each language. This can be quite limited as it is not clear whether the approach can be well generalized to other styles.
- The evaluation task and the use case discussed in this work are also limited. The paper solely focuses on the stylized QA task. However, the effectiveness of style transfer or editing techniques should be proven under more realistic settings, such as conversation and more general user-AI interactions.
- The paper only applied Qwen-1.5-14B-Chat as the base LLM. It’s not clear whether the improvement and the conclusions can be generalized to other LLMs, such as LLaMA.

**Questions:**

- For the SFT baselines, have the authors also considered applying full finetuning? What would be the performance?
- Have the authors tried to apply the approaches to other sizes of LLMs? For example, what would be the results when you apply DRESS to a 7B/8B LLM?

---

> ### Author Response · Authors · 2024-11-23
> **Rebuttal to Reviewer jiFa (1)**
>
> We sincerely appreciate your comments, suggestions, and every effort spent on reviewing our work. Here we attempt to address all your remaining concerns. In the following, we quote your comments and then give our detailed response point-by-point.
>
> > **W1. GPT-4 reliance of our data construction process**
>
> Nowadays, as the capability of GPT model family increases, it has gradually been a **common practice** to create or augment synthetic data generated by strong closed-source LLMs to enhance the performance of smaller LLMs [1, 2]. In recent years, NeurIPS has also started the Dataset & Benchmark paper track, where plenty of published research [1] utilizes closed-source LLMs like GPT-4 for data synthesis and demonstrates outstanding data quality and downstream performances. However, there indeed exist some imperfect cases in GPT-4 generation (e.g., Semantics of stylized responses sometimes subtly deviate from the original ones in our pipeline), yet we have proposed a strategy in Sec. 4.3 to filter out those imperfect semantic noises and edit the style-relevant subspace only. In other words, **our method is robust to the limitations of GPT-4**. Hence, to summarize, distilling knowledge from GPT-4-generated synthetic data has been a common practice to enhance LLM performance, and the underlying limitations can be mitigated in our method.
>
> [1] Toshniwal S, Moshkov I, Narenthiran S, et al. OpenMathInstruct-1: A 1.8 million math instruction tuning dataset[C]. NeurIPS 2024 Dataset & Benchmark Track (Oral).
>
> [2] Ge T, Chan X, Wang X, et al (Tencent AI Lab (Seattle)). Scaling Synthetic Data Creation with 1,000,000,000 Personas[J]. arXiv 2024
>
> > **W2. Generalizability to other styles**
>
> In our work, we selected two representative language styles from the most widely used and the most natively spoken languages (i.e., English and Chinese). Both are from an earlier era, where there are much **fewer corpora of the same style that could be inherently learned in the pretrained LLMs**. Hence, those two benchmarks are very challenging since LLMs do **NOT** have sufficient inherent knowledge to imitate the style **solely with prompts**. Compared with other types of styles (e.g., emotions, tones such as humorous), LLMs can easily adapt based on the data they have encountered, which does **NOT** pose a significant challenge. Moreover, adapting to our selected styles requires adjustments in multiple aspects, including differences in vocabulary between early modern and contemporary language, sentence construction logic, rhetorical techniques, and whether expressions are implicit or explicit, etc. These adaptations demand a very high level of abstraction and attention to detail to achieve proper alignment. In conclusion, our constructed benchmark is **comprehensive, representative, generalizable and challenging**, which is one of the core contributions of this paper.
>
> > **W3. Have we considered more realistic conversational eval setting?**
>
> We suspect that there might be some misunderstanding of our “QA” task. This work, as introduced in the Intro, is inspired from the real-world challenges of building interactive or humanoid conversational agents, especially for NPC construction and character roleplaying. Hence, we actually evaluated QA tasks comprising **several different types of real-world question-answering scenarios**, such as **daily conversations, commonsense QA and plot-specified roleplaying, etc.**, which **covers most scenarios that the reviewer mentioned**. Here we supplement some tested cases as below.
>
> Case 1: Dream of the Red Chamber Style
> ```
> Query: 我今天心情不好，昨晚没睡好。
> [DRESS Response] 瞧见你今日脸色不太好，昨晚定是没能安歇。要不咱找个清静地方，喝口热茶，舒舒筋骨？心情不好，憋在心里可不痛快，有啥心事不妨掏出来让我听听，兴许能帮你散散晦气。
> [Original Response] 听到你今天心情不太好，我感到很遗憾。希望你能找个安静的地方休息一下，或者做些你喜欢的事情来放松心情。睡眠对情绪有很大影响，如果昨晚确实没睡好，可以考虑午休片刻，或者调整一下作息时间，保证充足的睡眠对改善心情很有帮助。如果你感到烦恼，欢迎你随时向我倾诉，我们可以一起探讨和解决问题。
> ```
> Case 2: Shakespeare Style
> ```
> [Query] I spotted that chocolate you like at the supermarket just now, so I picked some up for you.
> [DRESS Response] Fair friend, thy kindness doth my heart delight, as sweet as honeyed nectar to my soul's sight. Thy thoughtful act, a token of thy love doth bring, A joy that doth my senses in sweet rapture cling.
> [Original Response] Aw, thank you so much! You're too kind! I'm really looking forward to indulging in that delicious chocolate. You always know how to make me happy!
> ```
> Those cases all show that DRESS can make LLMs obtain great conversational capability with a very expressive and precise style imitation.

---

> ### Author Response · Authors · 2024-11-23
> **Rebuttal to Reviewer jiFa (2)**
>
> > **W4/Q2. Generalizability to other base models (or smaller size models)**
>
> We appreciate your suggestion and supplement the experiments on LLaMA-3-8B as the base model. The results are shown below.
>
> **[Dream of the Red Chamber style]**
>
> | LLaMA-3-8B    | SI   | SP   | FS   | OA   | GPT4  |
> |---------------|------|------|------|------|-------|
> | 3-shot Prompt | 38.8 | 71.4 | 38.1 | 10.6 | 5.44  |
> | ITI           | 82.0 | 68.1 | 37.9 | 21.2 | 7.25  |
> | DRESS         | 85.3 | 71.1 | 41.4 | 25.1 | 7.53  |
>
> **[Shakespeare style]**
>
> | LLaMA-3-8B    | SI   | SP   | FS   | OA   | GPT4  |
> |---------------|------|------|------|------|-------|
> | 3-shot Prompt | 99.8 | 69.1 | 40.6 | 28.0 | 8.85  |
> | ITI           | 99.5 | 73.2 | 43.1 | 31.4 | 9.08  |
> | DRESS         | 100  | 75.0 | 43.7 | 32.7 | 9.14  |
>
> Results demonstrate that our methods still consistently perform well on LLaMA-3-8B. Especially for Dream of the Red Chamber benchmark, it is observed that prompting method cannot conduct successful stylization probably due to the lack of pretraining corpora of this style and even **fails to imitate few-shot samples**. While our method significantly outperforms the SOTA baselines and achieve a better stylization quality. The results are supplemented in Appendix B of our updated manuscript.
>
> > **Q1. Have we considered Full Fine-tuning methods for SFT baseline?**
>
> We actually considered using Full Fine-tuning (FFT) for SFT, yet PEFT methods such as LoRA are a more **economic and effective** choice for style adaptation. The reason is that for style adaptation, we often do not have a large amount of training data, and the **full fine-tuning strategy may lead to overfitting** (e.g., learning to respond with some specific tokens instead of learning what the style really depicts). Research works like [3, 4] also demonstrate that FFT suffers from overfitting and cannot perform as good as other PEFT methods. Moreover, the **training cost of FFT is relatively large** for our experimental hardware setting (8x24GB GPUs), where we have to sacrifice the floating-point precision to start the training (using full precision will lead to GPU memory overflow), and moreover, training efficiency could be a bottleneck. We also supplement the FFT results on the Shakespeare benchmark below.
>
> | LLaMA-3-8B       | SI   | SP   | FS   | OA    |
> |------------------|------|------|------|-------|
> | SFT (LoRA)       | 95.5 | 69.8 | 36.8 | 24.5  |
> | SFT (Full, bf16) | 95.0 | 70.2 | 36.4 | 24.3  |
>
> Results demonstrate that the performance gap between LoRA and Full Fine-tuning is not significant. Therefore, taking both training efficiency and effectiveness into consideration, we choose to apply SOTA PEFT method LoRA as the optimization algorithm for SFT.
>
> [3] Liu W, Qiu Z, Feng Y, et al. Parameter-efficient orthogonal finetuning via butterfly factorization[C]. ICLR 2024
>
> [4] Zhang Q, Chen M, Bukharin A, et al. AdaLoRA: Adaptive budget allocation for parameter-efficient fine-tuning[C]. ICLR 2023.

---

> > ### Comment · Reviewer_jiFa · 2024-11-25
> >
> > Thank you for the detailed responses. I would raise my score to 6.

---

> > > ### Author Response · Authors · 2024-11-26
> > >
> > > Thank you for your time, effort, and positive assessment. Your guidance and insights have been invaluable in refining our research. We remain committed to addressing any remaining concerns and ensuring a better quality of our work.

---

### Author Response · Authors · 2024-11-23
**General Rebuttal**

Dear ICLR PCs, ACs and all reviewers,

We are pleased to hear all the suggestions and comments on our work. There have been supporting voices about the significance of our proposed research issue, the efficiency, effectiveness, and soundness of our proposed methods, the easy-to-follow and well-structured writing, and positive overall comments like "interesting", "convincing", and "well-motivated". There are also several questions and modification suggestions, and we have responded to them point-by-point in the rebuttal to each reviewer. Here we summarize some main concerns raised by the reviewers and provide an overall explanation to each of them.

- [jiFa] **Errors may be incorporated during the data synthesis process**: We have designed denoising modules (i.e., attention head filtering and style subspace filtering) to mitigate this problem. The results also demonstrate the effectiveness of the design.
- [jiFa, DDuX, mVfE] **The generalizability of our proposed benchmark dataset**: This dataset comprises two distinct language styles (Shakespeare-style in English, and Dream of the Red Chamber-style in Chinese). Each of them is challenging, as they are several hundred years from now, without enough corpora for LLMs to pre-train and learn inherently. Moreover, they are more comprehensive, as each of them requires a very comprehensive imitation of the sentence structure, vocabulary, tones, rhetorical techniques, expressions, etc. Therefore, a well-performed method on our benchmark should be easily generalized to most scenarios, which demonstrates the generalizability of our proposed benchmark dataset.
- [jiFa, mVfE] **The generalizability to other base models in other sizes**: We have supplemented the experiments on LLaMA-3-8B to validate this point.
- [DDuX] **More ablation studies**: We have explained what we have done originally and supplemented some lacking experiments and explanations in our rebuttal.
- [AsRZ] **Applicability under data-hunger scenario**: We regard this as a very constructive suggestion, and we have done some experiments on this. The results show that our method also performs well when the data is scarce, better than the prompting methods.
- [mVfE] **Concerns about the difference between our method and prompting methods**: We claim that data synthesis based on prompting closed-source LLMs should NOT be regarded as "prompt-based" methods (e.g., using prompted synthetic data for self-play or tuning is NOT a prompt-based method). The core contribution is how we utilize the knowledge underlying those synthetic data.
- [mVfE] **Notation problems**: We have corrected them in our new manuscript.

We have also modified our manuscript accordingly, where the changes are highlighted in red.

Thanks again to all the reviewers for all the efforts made in reviewing our paper, and we genuinely hope the reviewers could consider a more positive evaluation of our work.

---

### Meta-Review · Area_Chair_isUB · 2024-12-23

**Metareview:**

This paper presents DRESS (Disentangling Representation Editing in Style Subspace), a novel approach for generating stylized LLM responses through representation editing. The method leverages style-relevant subspaces within the model's representation space to conduct targeted editing while preserving semantic meaning. Initial reviewer concerns focused on the reliance on GPT-4 for data synthesis, limited style types tested, and questions about the method's relationship to prompting approaches. The authors addressed these by demonstrating DRESS's effectiveness with different base models, showing strong performance even with limited training data, and clarifying how their representation editing approach fundamentally differs from prompt-based methods through comprehensive ablation studies.

Post-rebuttal discussions revealed general agreement on the technical soundness and novelty of the approach, though some concerns remained about broader applicability. The AC agrees that the paper doesn't fully address persona creation and personality aspects of AI agents, but it is an important foundational step toward that goal. The systematic evaluation on challenging literary styles and demonstrated improvements over baselines, even in low-resource settings, support the paper's value as a cornerstone contribution toward more sophisticated AI agent development. The AC believes this work is a solid contribution to the research community and worth an acceptance.

**Additional Comments On Reviewer Discussion:**

Post-rebuttal discussions revealed general agreement on the technical soundness and novelty of the approach, though some concerns remained about broader applicability. The AC agrees that the paper doesn't fully address persona creation and personality aspects of AI agents, but it is an important foundational step toward that goal.

---

### Decision · Program_Chairs · 2025-01-22

Accept (Poster)